# Large-scale features and evaluation of the PMIP4-CMIP6 *midHolocene* simulations

Chris M. Brierley[1], Anni Zhao[1], Sandy P. Harrison[2], Pascale Braconnot[3], Charles J. R. Williams[4,5], David J. R. Thornalley[1], Xiaoxu Shi[6], Jean-Yves Peterschmitt[3], Rumi Ohgaito[7], Darrell S. Kaufman[8], Masa Kageyama[3], Julia C. Hargreaves[9], Michael P. Erb[8], Julien Emile-Geay[10], Roberta D'Agostino[11], Deepak Chandan[12], Matthieu Carré[13,14], Partrick J. Bartlein[15], Weipeng Zheng[16], Zhongshi Zhang[17], Qiong Zhang[18], Hu Yang[6], Evgeny M. Volodin[19], Robert A. Tomas[20], Cody Routson[8], W. Richard Peltier[12], Bette Otto-Bliesner[20], Polina A. Morozova[21], Nicholas P. McKay[8], Gerrit Lohmann[6], Allegra N. Legrande[22], Chuncheng Guo[17], Jian Cao[23], Esther Brady[20], James D. Annan[9], and Ayako Abe-Ouchi[7,24]

[1]Department of Geography, University College London, London, WC1E 6BT, UK
[2]Department of Geography and Environmental Science, University of Reading, Reading, RG6 6AB, UK
[3]Laboratoire des Sciences du Climat et de l'Environnement-IPSL, Unité Mixte CEA-CNRS-UVSQ, Université Paris-Saclay, Orme des Merisiers, Gif-sur-Yvette, France
[4]Department of Meteorology, University of Reading, Reading, RG6 6BB, UK
[5]School of Geographical Sciences, University of Bristol, Bristol, BS8 1SS, UK
[6]Alfred-Wegener-Institut Helmholtz-Zentrum für Polar- und Meeresforschung, Bremerhaven, Germany
[7]Japan Agency for Marine-Earth Science and Technology, Yokohama, Japan
[8]School of Earth and Sustainability, Northern Arizona University, Flagstaff, AZ 86011, USA.
[9]Blue Skies Research Ltd, Settle, BD24 9HE, UK
[10]Department of Earth Sciences, University of Southern California, Los Angeles, California, USA
[11]Max Planck Institute for Meteorology, Hamburg, Germany
[12]Department of Physics, University of Toronto, Ontario, M5S1A7, Canada
[13]LOCEAN Laboratory, Sorbonne Universités (UPMC, Univ Paris 06)-CNRS-IRD-MNHN, Paris, France
[14]CIDIS-LID-Facultad de Ciencias y Filosofía-Universidad Peruana Cayetano Heredia, Lima, Peru
[15]Department of Geography, University of Oregon, Eugene, OR 97403, USA
[16]LASG, Institute of Atmospheric Physics, Chinese Academy of Sciences, Beijing 100029, China
[17]NORCE Norwegian Research Centre, Bjerknes Center for Climate Research, Bergen, Norway
[18]Department of Physical Geography and Bolin Centre for Climate Research, Stockholm University, 10691, Stockholm, Sweden
[19]Marchuk Institute of Numerical Mathematics, Russian Academy of Sciences, ul. Gubkina 8, Moscow, 119333, Russia
[20]Climate and Global Dynamics Laboratory, National Center for Atmospheric Research (NCAR), Boulder, CO 80305, USA
[21]Institute of Geography, Russian Academy of Sciences, Staromonetny L. 29, Moscow, 119017, Russia
[22]NASA Goddard Institute for Space Studies, New York, NY 10025, USA
[23]School of Atmospheric Sciences, Nanjing University of Information Science & Technology Nanjing, 210044, China
[24]Atmospheric and Ocean Research Institute, The University of Tokyo, Kashiwa, Japan

**Correspondence:** c.brierley@ucl.ac.uk

**Abstract.** The mid-Holocene (6,000 years ago) is a standard time period for the evaluation of the simulated response of global climate models using paleoclimate reconstructions. The latest mid-Holocene simulations are a paleoclimate entry card for the Palaeoclimate Model Intercomparison Project (PMIP4) component of the current phase of the Coupled Model Intercomparison

Project (CMIP6). Here we provide an initial analysis and evaluation of the results of the experiment for the mid-Holocene. We show that state-of-the-art models produce climate changes that are broadly consistent with theory and observations, including increased summer warming of the northern hemisphere and associated shifts in tropical rainfall. Many features of the PMIP4-CMIP6 simulations were present in the previous generation (PMIP3-CMIP5) of simulations. The PMIP4-CMIP6 ensemble for the mid-Holocene has a global mean temperature change of -0.3 K, which is -0.2 K cooler than the PMIP3-CMIP5 simulations predominantly as a result of the prescription of realistic greenhouse gas concentrations in PMIP4-CMIP6. Biases in the magnitude and the sign of regional responses identified in PMIP3-CMIP5, such as the amplification of the northern African monsoon, precipitation changes over Europe and simulated aridity in mid-Eurasia, are still present in the PMIP4-CMIP6 simulations. Despite these issues, PMIP4-CMIP6 and the mid-Holocene provide an opportunity both for quantitative evaluation and derivation of emergent constraints on the hydrological cycle, feedback strength and potentially climate sensitivity.

## 1 Introduction

Future climate changes pose a major challenge for Human civilisation, yet uncertainty remains about the nature of those changes. This arises from societal decisions about future emissions, internal variability, and also uncertainty stemming from differences between the models used to make the projections (Hawkins and Sutton, 2011; Collins et al., 2013). Coupled general circulation models (GCMs) can be used to simulate past changes in climate as well as those of the future. Palaeoclimate simulations allow us to test the theoretical response of such models to various external forcings and provide an independent evaluation of them. The Coupled Model Intercomparison Project (CMIP; Eyring et al., 2016), which coordinates efforts to compare climate model simulations, includes simulations designed to test model performance under past climate regimes. Evaluation of these palaeoclimate simulations against palaeoclimate reconstructions, coordinated through the Palaeoclimate Modelling Intercomparion Project (PMIP; Kageyama et al., 2018), provides an independent test of the ability of state-of-the-art models to simulate climate change.

The mid-Holocene (6000 years ago, 6ka) is one of the palaeoclimate simulations included in the current phase of CMIP (PMIP4-CMIP6; Otto-Bliesner et al., 2017). This period is characterised by an altered seasonal and latitudinal distribution of incoming solar radiation, because of larger obliquity and orbital precession, meaning that the Earth was closest to the Sun in boreal autumn (rather than in boreal winter as today) and that the northern latitudes received more solar radiation than today. The mid-Holocene has been a baseline experiment for PMIP since its inception (Joussaume et al., 1999; Braconnot et al., 2007, 2012). As such, it has been a focus for synthesis of palaeoenvironmental data (see summary in Harrison et al., 2016) and for the reconstruction of palaeoclimate variables from these data (e.g. Kohfeld and Harrison, 2000; Bartlein et al., 2011) to facilitate systematic model evaluation (e.g. Hargreaves et al., 2013; Jiang et al., 2013; Prado et al., 2013; Harrison et al., 2014; Mauri et al., 2014; Perez-Sanz et al., 2014; Harrison et al., 2015; Bartlein et al., 2017).

The PMIP4-CMIP6 simulations differ from previous palaeoclimate simulations in two ways. Firstly, they represent a new generation of climate models with greater complexity, improved parameterisations and often run at higher resolution. Changes to the model configuration have, in some cases (e.g. CCSM4/CESM2, HadGEM2/HadGEM3, IPSL-CM5A/IPSL-CM6A),

resulted in substantially higher climate sensitivity than the previous PMIP3-CMIP5 version of the same model, although this is not a feature of all of the models (Tab. 1,2). Preliminary investigations suggest point at stronger cloud feedbacks as the cause (Zelinka et al., 2020), which may also influence the model sensitivity to the mid-Holocene external forcing. Secondly, the protocol for the PMIP4-CMIP6 mid-Holocene experiment (called *midHolocene* on the Earth System Grid Federation, and henceforth herein) was designed to represent the observed forcings better than in previous mid-Holocene simulations (Otto-Bliesner et al., 2017). In addition to the change in orbital configuration, which was the only change imposed in the PMIP3-CMIP5 experiments, the current experiments include a realistic specification of atmospheric greenhouse gas concentrations. Because of the lower values of greenhouse gas concentrations, the PMIP4-CMIP6 simulations are expected to be slightly colder than the PMIP3-CMIP5 experiments (Otto-Bliesner et al., 2017). The model configuration and all other forcings are the same as in the pre-industrial control simulation (*piControl*, 1850 CE). This means that models with dynamic vegetation in the *piControl* are run with dynamic vegetation in the *midHolocene* experiment, so the PMIP4-CMIP6 ensemble includes a mixture of simulations with prescribed or interactive vegetation. Although some of the models were run with an interactive carbon cycle, none included fully-dynamic vegetation.

Here, we provide a preliminary analysis of the PMIP4-CMIP6 *midHolocene* simulations, focusing on surface temperature changes (sec. 3.1), hydrological changes (sec. 3.2 & 3.3) and the deep ocean circulation (sec. 3.4). We examine the impact of changes in model configuration and experimental protocol on these simulations, specifically how far these changes improve known biases in the simulated changes. We draw on an extended set of observation-derived benchmarks to evaluate these simulations. Finally we discuss the implications of this evaluation for future climate changes, including investigating whether changes in climate sensitivities between the CMIP6 and CMIP5 models has an impact on the simulations.

## 2 Methods

### 2.1 Experimental Setup and Models

The protocol and experimental design for the PMIP4-CMIP6 *midHolocene* simulations are described by Otto-Bliesner et al. (2017) and Earth System Documentation (2019). The *midHolocene* simulations are run with known orbital parameters for 6000 yr BP and atmospheric trace greenhouse gas concentrations (GHGs) derived from ice-core records (as described by Otto-Bliesner et al., 2017). Eccentricity is increased by 0.001918 in the *midHolocene* simulations relative to the *piControl*, obliquity is increased by 0.646°, and perihelion ($\omega$ - 180°) is changed from 100.33°in the *piControl* (in January) to 0.87°in the *midHolocene* (near the boreal autumn equinox). The result of these astronomical changes is a difference in the seasonal and latitudinal distribution of top-of-atmosphere (TOA) insolation. During boreal summer, insolation between 40-50°N was 25 $W/m^2$ higher in the *midHolocene* simulations than in the *piControl* (Otto-Bliesner et al., 2017). The long-lived greenhouse gases are specified at their observed concentrations. Carbon dioxide is specified at 264.4 ppm (vs 284.3 ppm during the pre-industrial) and methane at 597 ppb (vs 808 ppb) and $N_2O$ at 262 ppb (versus 273 ppb). These changes in GHG concentrations lead to an effective radiative forcing of -0.3 $W/m^2$ (Otto-Bliesner et al., 2017).

Sixteen models (Tab. 1) have performed the PMIP4-CMIP6 *midHolocene* simulations. A similar number of models have performed the equivalent PMIP3-CMIP5 *midHolocene* simulation (Tab. 2). The PMIP4-CMIP6 simulations are either available from the Earth System Grid Federation (from which they are freely downloadable) or will be lodged there in the near future. We evaluate these simulations as part of an ensemble and only sometimes identify individual models. Most of the models included in the PMIP4-CMIP6 ensemble are state-of-the-art climate models, but we also include some results from models that are either lower resolution or less complex (and therefore faster). Even though all models have the same orbital parameters and trace gases in the *midholocene* experiment, differences in the specification of other boundary conditions can mean that the forcing is not identical in every model. For example, the models may have slightly varying solar constants (see notes in Table 1), reflecting choices made by the different groups for the *piControl* simulations. Similarly, the orbital parameters used by some groups for the *piControl* are the same as for the historical simulation and the trace gases are slightly different from the PMIP4-CMIP6 recommendations. Differences in the pre-industrial planetary albedo, resulting from surface albedo and clouds, may also mean the effective solar forcing is different between models (Braconnot et al., 2012). Experimental setup and spin-up procedure are documented for each *midHolocene* simulation individually elsewhere (following the recommendation of Otto-Bliesner et al., 2017).

## 2.2 Analysis techniques and calendar adjustments

Fixed monsoon domains are often used when investigating variability and future changes in monsoon rainfall (e.g. Christensen et al., 2013). However, this is not appropriate in the mid-Holocene when the monsoons were greatly extended. Following Jiang et al. (2015), we adopt the definition of Wang et al. (2011) for analysis of monsoon regions: a grid point is considered to be affected by the monsoon if the rainfall predominantly falls in the summer (May–Sept. in the Northern Hemisphere, Nov.– March in the Southern Hemisphere; assessed using summer rainfall forming at least 55% of the annual total) and the average rain rate difference between summer and winter (called monsoon intensity) is at least 2 mm/day or more. The ensemble mean global domain is determined by applying both these criteria to the ensemble mean summer rainfall and monsoon intensity. We calculate annual (November-October) times series of the areal extent for 7 land-based monsoon systems (Christensen et al., 2013), as well as determining the average precipitation rate within each system. Internal climate variability is characterised by the standard deviation of these annual time series. The integral of these values is the total monsoon rainfall for each regional monsoon.

The *midHolocene* experiment involves redistributing the incoming insolation spatially and through the year (Otto-Bliesner et al., 2017). This altered orbital configuration during the mid-Holocene resulted in a change in the Earth's transit speed along different parts of its orbit such that, when considered as angular fractions of the Earth's orbit, the month lengths differed during the mid-Holocene (Joussaume and Braconnot, 1997; Bartlein and Shafer, 2019). Northern Hemisphere winter (December, January, February; DJF) was longer and summer (June, July, August; JJA) correspondingly shorter from an insolation perspective than in the present day and the *piControl* simulation. However simulation output by CMIP6 models is restricted to modern calendars (Juckes et al., 2019). This is not a problem for annual or daily diagnostics, but summarizing model output using only the modern calendar prohibits straightforward adjustment of the numbers of days over which the aggregation of monthly

simulation output takes place. To take account of these differences in calculating monthly or seasonal variables, we use the PaleoCalAdjust software (Bartlein and Shafer, 2019), which interpolates from non-adjusted monthly averages to pseudo-daily

values and then calculates the average values for adjusted months defined as angular fractions of the orbit. This software has been favourably evaluated for monthly temperature and precipitation variations with both PMIP4-CMIP6 and transient simulations (Bartlein and Shafer, 2019). Given the experimental protocol fixes the date of the vernal equinox as 21st March (Otto-Bliesner et al., 2017), the largest impact of the calendar adjustment occurs in September (a key month for Arctic sea ice coverage). The PaleoCalAdjust software computes adjusted monthly variables from original monthly means, a computation

which could impact the accuracy of variables that change abruptly throughout the year, rather than gradually, such as the sudden increase in precipitation in monsoon regions (Pollard and Reusch, 2002). To explore whether potential interpolation errors from PaleoCalAdjust are justified in such situations, we analysed the averaged rain rate during the monsoon season over the South American monsoon domain in the IPSL-CM6A-LR *midHolocene*, for which daily-resolution data is also provided on the Earth System Grid Federation. Since the areal extent of South American monsoon domain varies slightly when using different

temporal data, we make this comparison based on the grid points that always fall within the monsoon domain to provide the most robust assessment of the impact of the change in calendar. The average monsoon rain rate from the daily-resolution data is 7.0 mm/day: compared to 6.7 mm/day from calendar-adjusted monthly data and 7.1 mm/day using monthly data without this adjustment. The average monsoon rain rate in the *piControl* is 7.5 mm/day. We have therefore not applied the calendar adjustment when analysing monsoon variables.

The analysis presented here mainly uses generalised evaluation software tools derived from the Climate Variability Diagnostics Package (Phillips et al., 2014), which has been adapted for palaeoclimate purposes (Brierley and Wainer, 2018). It uses the surface air temperature and precipitation rate variables ('tas' and 'pr' respectively in the ESGF controlled vocabulary; Juckes et al., 2019), as well as several different ocean overturning mass streamfunction and sea ice concentration variables.

### 2.3  Palaeoclimate reconstructions and model evaluation

We provide only a preliminary quantitative evaluation of the realism of the PMIP4-CMIP6 simulations, drawing attention to obvious similarities and mismatches between the simulations and observational evidence of past climates. We concentrate our evaluation on two compilations of quantitative reconstructions from a number of sources. We use temperature reconstructions from the recent 'Temperature 12k' database (Kaufman et al., 2020b). We extracted anomalies for the mid-Holocene compared to the last millennium interval ($6.0 \pm 0.5ka - 0.6 \pm 0.5ka$) for site-level comparison with the PMIP4-CMIP6 simulations. This

database has 1319 time series reconstructions of temperature (mean annual, summer and winter temperature) based on a variety of different ecological, geochemical and biophysical marine (209) and terrestrial (470) sites (Kaufman et al., 2020b). Additionally, area-averaged temperature anomalies (w.r.t. 1800-1900) over 30° latitudinal bands have been generated using five different methods (Kaufman et al., 2020a) to yield a single composite value with confidence intervals. Bartlein et al. (2011) provide pollen-based reconstructions of land climate (mean annual temperature, mean temperature of the coldest month, grow-

ing season temperature, mean annual precipitation and the ratio of actual to potential evaporation), although we mainly focus on mean annual temperature and precipitation here. They combined the reconstructions at individual pollen sites to produce

an estimate for a 2°x2° grid (a resolution comparable with the climate models) reconstruction uncertainties are estimated as a pooled estimate of the standard errors of the original reconstructions for all sites in each grid cell. There is good coverage of northern hemisphere terrestrial sites, although there are gaps in the coverage especially for the tropics and southern hemisphere

(Bartlein et al., 2011). The Bartlein et al. (2011) data set was extended with some speleothem and ice core temperature reconstructions and used to evaluate the PMIP3-CMIP5 simulations (Harrison et al., 2014). In this study we use the pollen-only data set from Bartlein et al. (2011) and the multi-proxy data set (Kaufman et al., 2020b) to provide a measure of the uncertainties in reconstructed climates, although differences in methodology and coverage preclude direct comparison between the two data sets. We incorporate an additional data set to facilitate comparisons of the northern African monsoon between

the CMIP6-PMIP4 simulations and previous generations of simulations, namely water-balance estimates of the quantitative change in precipitation required to support the observed mid-Holocene vegetation change at each latitude compared to present (Joussaume et al., 1999).

## 3  Simulated mid-Holocene Climates

### 3.1  Temperature Response

As expected from the insolation forcing, the PMIP4-CMIP6 ensemble shows an increase in mean annual temperature (MAT) as compared to *piControl* conditions in the high northern and southern latitudes and over Europe (Fig. 1a). Yet there is a decrease in MAT elsewhere, which is especially large over northern Africa and India. The ensemble produces a global cooling of -0.3°C compared to the *piControl* simulation (Tab. S2). The relatively small change in MAT is consistent with the fact that the *midHolocene* changes are largely driven by seasonal changes in insolation. The geographic patterns of temperature

changes in the PMIP4-CMIP6 ensemble are very similar to those seen in the PMIP3-CMIP5 ensemble. However, the change in MAT with respect to the *piControl* in the PMIP4-CMIP6 ensemble is generally cooler than in the PMIP3-CMIP5 (Fig. 1). The difference in the experimental protocol between the two sets of simulations would be expected to cause a slight cooling, since the difference in GHG concentrations results in an effective radiative forcing of around -0.3 $W/m^{-2}$ (Otto-Bliesner et al., 2017). To evaluate this, we estimate the ensemble-mean forced response (Fig. 1f) based on the climate sensitivity of

each model (Tab. 1) and pattern scaling. The estimated global mean pattern-scaled anomaly is -0.28°C, roughly similar to the difference between the two model generations (Fig. 1, Tab. S2).

In-line with theory, the higher insolation in northern hemisphere (NH) summer results in a pronounced summer (JJA) warming, particularly over land (Fig. 2). The increase in summer temperature over land in the NH high latitudes in the ensemble mean is 1.1°C (Tab. S2). Increased NH summer insolation leads to a northward shift and intensification of the monsoons (sec.

3.2), with an accompanying JJA cooling in the monsoon-affected regions of northern Africa and and South Asia. Reduced insolation in the NH winter (DJF) results in cooling over the northern continents and this cooling extends into the northern tropical regions, although the Arctic is warmer than in the *piControl* simulation (Fig. 2). Although the Southern Ocean shows warmer temperatures in the *midHolocene* than the *piControl* simulations in austral summer (DJF) as a result of increased obliquity, this warming does not persist into the winter to the same extent as seen in the Arctic. The damped insolation seasonality,

together with the large effective heat capacity of the ocean heavily damps seasonal variations in surface air temperature in the Southern Ocean. The enhanced NH seasonality and the preponderance of land in the NH cause seasonal variations of the inter-hemispheric temperature gradient, which results in a small warming of the northern hemisphere at the expense of the southern hemisphere in the annual, ensemble mean. The PMIP4-CMIP6 ensemble is cooler than the PMIP3-CMIP5 ensemble in both summer and winter (Fig. 2). The pattern of cooling in both seasons is very similar (not shown) to the annual mean ensemble difference in Fig. 1e, further supporting the lower greenhouse gas concentrations in the experimental protocol (sect. 2.1) as the cause of the cooling.

Biases in the control simulation may influence the response to mid-Holocene forcing (Braconnot et al., 2012; Ohgaito and Abe-Ouchi, 2009; Harrison et al., 2014; Braconnot and Kageyama, 2015) and certainly affect the pattern and magnitude of simulated changes. There is some difficulty in diagnosing biases in the *piControl*, because there are few spatially-explicit observations for the pre-industrial period, especially for precipitation. We therefore evaluate these simulations using reanalysed climatological temperatures (between 1871-1900 CE; Compo et al., 2011) for the spatial pattern (Fig. 3) and zonal averages of observed temperature (Fig. 4) for the period 1850-1900 CE from the HadCRUT4 dataset (Morice et al., 2012; Ilyas et al., 2017). We compare these with the mean difference between the pre-industrial climatology of each model (i.e. the ensemble mean bias). The PMIP4-CMIP6 models are generally cooler than the observations, most noticeably in polar regions, over land and over the NH oceans (Fig. 4). The models are too warm over the eastern boundary upwelling currents, although it remains to be seen whether this indicates improved representation of the relevant physical processes compared to PMIP3-CMIP5. The colder conditions over the Labrador Sea (Fig. 3b) also indicate difficulty with resolving the regional ocean circulation sufficiently. The polar regions are noticeably too cold in the ensemble mean (Fig. 3), but there is considerable spread between individual models (Fig. 4). There is no simple relationship between a model's representation of the pre-industrial temperature and the magnitude of its simulated mid-Holocene temperature response (Fig. 4). Other factors affect the regional direct and indirect response to mid-Holocene forcing, such as ice albedo and ocean temperature advection into the Arctic. PMIP4-CMIP6 also includes simulations with dynamic vegetation, for example. The associated vegetation-snow albedo feedback would tend to reduce the simulated cooling (e.g. O'ishi and Abe-Ouchi, 2011), but can introduce a larger cooling bias in the *piControl* simulation (Braconnot et al., 2019). However, changes in the treatment of aerosols in the PMIP4-CMIP6 ensemble could enhance the simulated cooling (Pausata et al., 2016; Hopcroft and Valdes, 2019).

Kaufman et al. (2020a) suggest that zonal, annual mean temperatures during the mid-Holocene were warmer at most latitudes (Fig. 4), with maximum warming in the Arctic, using the reconstructions in the Temperature 12k compilation (Kaufman et al., 2020b). Individual records in the Bartlein et al. (2011) compilation demonstrate the heterogeneity within these estimates (Fig. 4). The PMIP4-CMIP6 ensemble is equivocal about whether the polar regions were warmer or cooler on the annual mean. Furthermore, the PMIP4-CMIP6 models show a consistent cooling in the tropics. Tropical cooling was present, but less pronounced, in the PMIP3-CMIP5 ensemble (Fig. 4). Tropical cooling is not consistent with the Temperature 12k area-averages (Kaufman et al., 2020a) (although the Bartlein et al. (2011) compilation does not discount it, the majority of their reconstructions are solely from Africa). Interestingly comparisons over Europe and North America, both well-sampled by the Bartlein et al. (2011) compilation, the models appear to show too much warming in both summer and winter (Fig. S3).

205 Further work is required to determine whether the discrepancies between the temperature reconstructions and PMIP4-CMIP6 simulations arise from systematic model error, sampling biases in the data compilation (e.g. Liu et al., 2014b; Marsicek et al., 2018; Rodriguez et al., 2019) or a contribution from both sources.

There is substantial disagreement within the PMIP4-CMIP6 ensemble about the magnitude of the surface temperature changes at the regional scale. The intermodel spread of the temperature response across the PMIP4-CMIP6 ensemble is of 210 the same magnitude as the ensemble mean for both annual (Fig. 1) and seasonal (Fig. 2) temperature changes. There is a very large spread in the high-latitude oceans and adjacent land areas in the winter hemisphere, where the spread originates from inter-model differences in the extent of the simulated sea ice (sect. 3.4). Ice-albedo feedback would enhance inter-model temperature differences (Berger et al., 2013). The second region characterised by large inter-model differences is where there are large changes in precipitation in the tropics. This suggests that the spread originates in inter-model differences in simulated 215 large scale water advection, evaporative cooling, cloud cover and precipitation changes. There is no systematic reduction in the spread of temperature responses within PMIP4-CMIP6 ensemble compared to the PMIP3-CMIP5 ensemble (Fig. 1, Fig. 2). Each of the ensembles include models of different complexity, and the lack of a systematic difference suggest that complexity and model tuning has a larger impact on the responses than differences in the protocol. Thus, even though there is a protocol-forced cooling of PMIP4-CMIP6 relative to PMIP3-CMIP5, these simulations can be considered as subsets of a 220 single ensemble (see sect. 3.5; Harrison et al., 2014). However, given the large inter-model range in temperature changes in both subsets of this ensemble, it may be that classifying the models to highlight the impact of model complexity or of model biases on the climate response would be useful. This would also allow selection of subsets of the models for specific analyses, following a fit-for-purpose approach (Schmidt et al., 2014a).

### 3.2 Monsoonal Response

225 The enhancement of the global monsoon is the most important consequence of the mid-Holocene changes in seasonal insolation for the hydrological cycle (Jiang et al., 2015). The global monsoon domain is expanded in the PMIP4-CMIP6 *midHolocene* simulations: this occurs because of changes in both the summer rain rate and the monsoon intensity (Fig. 5). The weakening of the annual range of precipitation over the ocean and the strengthening over the continents indicates the changes reflect a redistribution of moisture (see e.g. Braconnot, 2004).

230 The most pronounced and robust changes in the monsoon occur over northern Africa and the Indian subcontinent (Fig. 6). The areal extent of the northern African monsoon is 20-50% larger than in the *piControl* simulations, but the average rain rate only increases by 10% (Fig. 7). The intensification of precipitation on the southern flank of the Himalayas (Table S2) in the *midHolocene* simulations is offset by a reduction in the Philippines and Southeast Asia (Fig. 6), so the area-averaged reduction in rain rate is reduced over the South Asian monsoon domain (Fig. 7). There is an extension and intensification of 235 the East Asian monsoon that is consistent across the PMIP4-CMIP6 ensemble, but the change is <10% (Fig. 7). Ensemble mean changes in the North American Monsoon System, and the Southern Hemisphere monsoons are also small (Fig. 6), and less consistent across the ensemble although most of the models show a weakening and contraction of the Southern American Monsoon System and Southern African monsoon (Fig. 7). Changes in internal climate variability within the monsoon systems

(characterised by standard deviations of the annual time series of both the areal extent and area-averaged rain rate; Fig. 7) are not consistent across the PMIP4-CMIP6 ensemble. Furthermore, those models that have the largest change in variability in one region are not necessarily the models that have large changes in other regions, which suggests that this variability is linked with regional feedbacks, rather than being an inherent characteristic of a model.

The broad scale changes in the PMIP4-CMIP6 simulations, with weaker southern and stronger/wider northern hemisphere monsoons, were present in the PMIP3-CMIP5 simulations (Fig. 6; testing the significance of the differences between the ensembles is discussed in sec. 3.5). The response is robust across model results, indicating that all models produce the same large scale redistribution of moisture by the atmospheric circulation in response to the interhemispheric and land-sea gradients induced by the insolation and trace gas forcing. At a regional scale, however, there are differences between the two ensembles. The PMIP4-CMIP6 *midHolocene* ensemble shows wetter conditions over the Indian Ocean, a larger northward shift of the ITCZ in the Atlantic and a widening of the Pacific rain belt compared to the PMIP3-CMIP5 models (Fig. 6). The expansion of the summer (JJA) monsoon in northern Africa is also greater in the PMIP4-CMIP6 than the PMIP3-CMIP5 ensemble (Table S2) and the location of the northern boundary is more consistent between models. This is associated with a better representation of the northern edge of the rainbelt for the *piControl* simulation in the PMIP4-CMIP6 ensemble compared with previous generations (Fig. S1). However, there is little relationship between the *piControl* precipitation biases and the simulated *midHolocene* changes in precipitation (Fig. S1). The variations in the *midHolocene* rainfall signal appear to be more related to monsoon dynamics rather than orbitally-induced local temperature variations (D'Agostino et al., 2019). The modulation of this dynamical response by the land surface and vegetation components of the PMIP4-CMIP6 models should be investigated.

Although the PMIP4-CMIP6 models show the expected expansion of the monsoons, this expansion is weaker than indicated by palaeoclimate reconstructions (Fig. 8 & S3). This was a feature of the PMIP3-CMIP5 simulations (Braconnot et al., 2012; Perez-Sanz et al., 2014) and previous generations of climate models (Joussaume et al., 1999; Braconnot et al., 2007). It has been suggested that this persistent mismatch between simulations and reconstructions arises from biases in the *piControl* (Harrison et al., 2015). Indeed, the ensemble mean global monsoon domain in the PMIP4-CMIP6 ensemble is more equatorward in the *piControl* compared to the observations, particularly over the ocean (Fig. 5). In northern Africa, the expansion of the monsoon domain in the *midHolocene* simulations merely removes the underestimation of its poleward extent in the *piControl* simulations (Fig. 5). Furthermore, evaluation of the *piControl* simulations using climatological precipitation data for the period between 1970 and the present day (Adler et al., 2003) shows the models fail to capture the magnitude of rainfall in the Intertropical Convergence Zone (ITCZ) and the southern portion of the South Pacific Convergence Zone (SPCZ). The SPCZ is too zonal because of the poor representation of the SST gradient between the Equator and 10°S in the west Pacific (Fig. 3; Brown et al., 2013; Saint-Lu et al., 2015). The PMIP4-CMIP6 models exhibit a dry bias over tropical and high northern latitude land areas, although the mid-latitude storm tracks are captured with varying levels of fidelity (Fig. 3).

There are large differences in the simulated change in mid-Holocene precipitation between different models, as shown by the standard deviation around the ensemble mean, in both the PMIP4-CMIP6 and PMIP3-CMIP5 ensembles (Fig. 6 & 8). Unsurprisingly, the largest differences between models occurs where the simulated change in precipitation is also largest (Fig. 6).

### 3.3 Extratropical hydrological responses

Hydrological changes in the extratropics are comparatively muted in the PMIP4-CMIP6 ensemble, and closely resemble features seen in the PMIP3-CMIP5 ensemble. There is a reduction in rainfall at the equatorward edge of the mid-latitude storm tracks, most noticeable over the ocean (Fig. 6). The NH extratropics are generally drier in the *midHolocene* simulations than in the *piControl*. There is a large inter-model spread in the summer rainfall changes over eastern North America and central Europe (Fig. 8). The spread in summer rainfall in both regions is clearly related to the large inter-model spread in summer temperature (c.f. Figs 2 & 6). Reconstructions from eastern North America suggest slightly drier conditions while reconstructions for central Europe show somewhat wetter conditions, but in neither case are these incompatible with the simulations.

There are regions, however, where there is a substantial mismatch between the PMIP4-CMIP6 simulations and the pollen-based reconstructions. There is a simulated reduction in summer rainfall in mid-continental Eurasia (Fig. 6). This reduction is somewhat larger in the PMIP4-CMIP6 ensemble than in the PMIP3-CMIP5 ensemble, although this difference is likely not significant (Fig. 8). However, this reduction in precipitation and the consequent increase in mid-continental temperatures is inconsistent with palaeoenvironmental evidence (and climate reconstructions), which show that this region was characterised by wetter and cooler conditions than today in the mid-Holocene (Fig. 8; Bartlein et al., 2017, Table S2). This indicates that model improvements have not resolved the persistent mismatch between simulated and observed mid-Holocene climate. Bartlein et al. (2017) pinpointed biases in the simulation of the extratropical atmospheric circulation as the underlying cause of this mismatch. The higher resolution of most PMIP4-CMIP6 models does not seem to improve the representation of these aspects of the circulation. Imperfect simulation of the extratropical circulation could also explain the failure to capture precipitation changes over Europe accurately (Mauri et al., 2014). The PMIP4-CMIP6 ensemble shows little change in mean annual precipitation over Europe (Fig. 6). Reconstructions of mid-Holocene precipitation suggest modest increases in northern Europe, increases in Central Europe, and much wetter conditions in the Mediterranean – something which is not captured by the PMIP4-CMIP6 ensemble (Fig. 8, Fig. S3).

### 3.4 Ocean and Cryospheric Changes

The AMOC is an important factor affecting the Northern Hemisphere climate system and is a major source of decadal and multidecadal climate variability (e.g. Rahmstorf, 2002; Lynch-Stieglitz, 2017; Jackson et al., 2015). Recent studies have reported a decline of up to ~15% in AMOC strength from the pre-industrial period to the present day (Rahmstorf et al., 2015; Dima and Lohmann, 2010; Caesar et al., 2018; Thornalley et al., 2018), at least partly in response to anthropogenic forcing. Reproducing the AMOC of the mid-Holocene is important for understanding the climate responses to external forcing at millennial timescales. The members of both the PMIP4-CMIP6 and PMIP3-CMIP5 ensemble have different AMOC strengths in their *piControl* simulations (Fig. 9), although all models correctly predict that it is stronger at 30°N than at 50°N. The PMIP4-CMIP6 models project a consistent reduction in AMOC under future scenarios (Weijer et al., 2020). There is a strong correlation (r=0.99 at 30°N) between the simulated strength of the AMOC in the *midHolocene* and the *piControl*. Furthermore, there is little change in the overall strength of the AMOC between the *midHolocene* and *piControl* experiments (Fig. 9) in either

the PMIP4-CMIP6 or the PMIP3-CMIP5 simulations, and no consistency in whether this comparatively small (and probably non-significant) change is positive or negative. Using a single metric to categorise changes in the AMOC is awkward – that two measures, both with their own uncertainties, indicate the same result increases our confidence that the overall changes were small. Shi and Lohmann (2016) detect large differences in simulated AMOC anomalies between models with coarse and higher resolutions. They suggest ocean and atmospheric processes affecting ocean salinity close to the sites of deep convection mean that higher resolution models tend to produce stronger *midHolocene* AMOC and lower resolution simulations a weaker AMOC than the *piControl*. The comparatively small changes in the AMOC strength between the PMIP4-CMIP6 *piControl* and *midHolocene* simulations are consistent with these earlier results, where the simulated changes are generally of less than 2 Sv (Fig. 9).

It is difficult to reconstruct past changes in the AMOC, especially its depth-integrated strength. Previous analyses have focussed on examining individual components of the AMOC, for example by using sediment grain size (Hoogakker et al., 2011; Thornalley et al., 2013; Moffa-Sanchez et al., 2015). The overall strength of the AMOC may be constrained by using sedimentary Pa/Th (e.g. McManus et al., 2004), although geochemical observations show that several additional factors influence Pa and Th distribution (Hayes et al 2013). The available Pa/Th records indicate no significant change in the AMOC between the mid-Holocene and the pre-industrial period (McManus et al., 2004; Ng et al., 2018; Lippold et al., 2019). Reconstruction of changes in the upper limb of the AMOC, based on geostrophic estimates of the Florida Straits surface flow, also indicate little change over the past 8000 years (Lynch-Stieglitz et al., 2009). Thus, overall, the palaeo-reconstructions are consistent with the simulated results (Fig. 9).

The altered distribution of incoming solar radiation at the mid-Holocene would be expected to alter the seasonal cycle of sea ice concentration. Analysis of simulations from previous generations of PMIP found a consistent reduction in Arctic summer sea ice extent at the mid-Holocene, and that the amount of sea ice reduction was related to the magnitude of warming in the region (Berger et al., 2013; Park et al., 2018). These findings hold for the PMIP4 models (Fig. 10). The PMIP4-CMIP6 models have slightly more realistic sensitivities of Arctic sea ice to warming and greenhouse gas forcing than PMIP3-CMIP5 models, but their simulated sea ice extents cover the same large spread easily encompassing the observations (SIMIP Community, 2020). There is little Arctic-wide relationship between the pre-industrial sea ice extent and its reduction at the mid-Holocene (Fig. 10). Local relationships may hold for key regions, such as the North Atlantic, where connections between pre-industrial sea ice coverage and mid-Holocene AMOC and summer sea ice reductions have been observed (Găinuşă-Bogdan et al., 2020). The changes in Arctic sea ice extent simulated for the *midHolocene* are generally amplified by the stronger insolation forcing imposed in the *lig127k* experiment (Otto-Bliesner et al., 2020b). Prior statistical analysis (Berger et al., 2013) supported by recent process-based understanding (SIMIP Community, 2020) suggests that further analysis of *midHolocene* sea ice changes would be informative for future Arctic projections (Yoshimori and Suzuki, 2019).

## 3.5 Evaluation of mid-Holocene climate features

Comparisons of the PMIP4-CMIP6 simulations with either palaeoenvironmental observations or palaeoclimate reconstructions have highlighted a number of regions where there are mismatches either in magnitude or sign of the simulated response. The

combination of the mismatches in, for example, simulated mean annual temperature or temperature seasonality results in an extremely poor overall assessment of the performance of each model (Fig. S2). This global assessment also provides little basis for discriminating between models, a necessary step in using the quality of specific midHolocene simulations operationally to enhance future projections for climate services (Schmidt et al., 2014a). At a regional scale (Fig. 4; Fig. 8; Fig. S3) it is clearly

possible to identify models that are unable to reproduce the observations satisfactorily. Thus, there would be utility in making quantitative assessment of model performance at a regional scale. Combining regional benchmarking of model performance with process diagnosis – to ensure that a model is correct because it captures the right processes – would therefore provide a firmer basis for using the *midHolocene* simulations to enhance our confidence in future projections.

    Analyses of key features of the *midHolocene* simulations, such as the monsoon amplification or the strength of the AMOC,

suggest that the PMIP4-CMIP6 simulations should regarded as from the same population as the PMIP3-CMIP5 simulations. We formally test this by calculating Hotelling's $T^2$ statistic (Wilks, 2011), a multivariate generalization of the ordinary $t$-statistic that is often used to examine differences in climate-model simulations (Chervin and Schenider, 1976), at each grid point of a common $1°$ grid for different combinations of climate variables. The patterns of "significant" (i.e. $p < 0.05$) tests (where one would reject the null hypothesis that the PMIP4-CMIP6 and PMIP3-CMIP6 ensemble means are equal) are random

(Fig. 11) and show little relation to the largest climate anomalies (Fig. 1 & 6). The total number of "significant" grid cells does not exceed the false discovery rate (Wilks, 2006). Consequently there is little support for the idea that the PMIP4-CMIP6 generation of simulations differ from the PMIP3-CMIP5 simulations, which were themselves not significantly different from the PMIP2-CMIP3 simulations (Harrison et al., 2015). This suggests, that all of these simulations could be considered as a single ensemble for process-based analysis (e.g. D'Agostino et al., 2019) or for the investigation of emergent constraints

(Yoshimori and Suzuki, 2019). Combining models from multiple ensembles could considerably enhance the statistical power of such analyses.

    Several of the PMIP4-CMIP6 models have a higher climate sensitivity, defined as the response of global temperature to a doubling of $CO_2$ (Gregory et al., 2004)), than earlier versions of the same model (Tab. 1, Tab. 2). This increased sensitivity could contribute to the PMIP4-CMIP6 ensemble being somewhat cooler than the PMIP3-CMIP5 ensemble. However, two

of the PMIP4-CMIP6 models have lower sensitivity and there is no real difference in the range of sensitivities of the two ensembles. This suggests that the change in the experimental protocol, specifically the fact that the specified atmospheric $CO_2$ concentration is ca 20 ppm lower in the PMIP4-CMIP6 experiments than in the PMIP3-CMIP5 experiments, is a more likely explanation for this change. This is borne out by comparison of the implied forcing as a result of the change in $CO_2$ (Fig. 1f) and the difference in temperature between the two ensembles (Fig. 1e).

There is no inherent relationship between climate sensitivity and seasonality, because the influence of the ocean is different on seasonal compared to multi-annual timescales. However, changes in climate sensitivity can arise from water vapour or cloud feedbacks, and thus it is feasible that changes in climate sensitivity could affect the simulated changes in seasonality. This is not borne out by analyses of seasonality changes in central Asia (Fig. 12): although four of the five individual models that have higher sensitivity in PMIP4-CMIP6 than the corresponding version of that model in PMIP3-CMIP5 show an increase

in the seasonality (Fig. 12), others do not support such a relationship. The fact that changes in climate sensitivity can be

detected in the thermodynamic response to orbital forcing, even though the relationship in this example is not constant, raises the possibility that the changes in seasonality shown in the *midHolocene* simulations could provide a constraint on climate sensitivity. Although we have not identified such a relationship in any region used to make model evaluations, analyses of other regions would help to verify this.

Circum-Pacific palaeoclimate records document marked fluctuations in ENSO activity throughout the Holocene (Tudhope et al., 2001; McGregor and Gagan, 2004; Koutavas and Joanides, 2012; McGregor et al., 2013; Cobb et al., 2013; Carré et al., 2014; Chen et al., 2016; Grothe et al., 2019). In the central and eastern Pacific, ENSO variability was reduced at 6 ka compared to present (Emile-Geay et al., 2016). This reduction has been simulated by models of various complexity (e.g. Clement et al., 2000; Liu et al., 2000; Zheng et al., 2008; Chiang et al., 2009; An and Choi, 2014; Liu et al., 2014a) and

is a feature of the PMIP4-CMIP6 *midHolocene* simulations (Table S2, Brown et al., submitted). Analyses of simulated and reconstructed changes in tropical Pacific climate variability (Emile-Geay et al., 2016) showed that the PMIP3-CMIP5 models rarely produced an ENSO as quiescent as shown by the palaeoclimate observations, though the imposition of mid-Holocene boundary conditions did increase those odds. This is also true for most of the PMIP4-CMIP6 models (Table S2). The models often produce a reduction in ENSO variability but, with the exception of MIROC-ES2L, this is much smaller than the reduction

implied by the palaeoclimate records. A key result of Emile-Geay et al. (2016) was that while PMIP3-CMIP5 models showed an inverse relationship between ENSO variance (inferred from 2-7yr bandpass filtered metrics of ENSO) and seasonality (defined as the range of the monthly-mean annual cycle), the observations showed either no relationship, or a weakly positive one. The analysis of the PMIP4-CMIP6 ensemble of Brown et al. (submitted) shows little to no relationship as well, in accordance with this set of palaeoclimate observations.

Palaeoenvironmental evidence also hints at an increased zonal SST gradient in the equatorial Pacific during the mid-Holocene (Koutavas et al., 2002; Linsley et al., 2010; Carré et al., 2014), whilst the PMIP4-CMIP6 ensemble yields a slight decrease in the gradient (Table S2). Analysis of equatorial Pacific climate change and variability finds little evidence for simulated relationship between SST gradient and ENSO variance in the PMIP4-CMIP6 ensemble (Brown et al., submitted).

## 4   Conclusions

The PMIP4-CMIP6 *midHolocene* simulations show changes in seasonal temperatures and precipitation that are in-line with the theoretical response to changes in insolation forcing. The broad-scale patterns of change are similar to those seen in previous generations of models, most particularly the PMIP3-CMIP5 ensemble. Both PMIP4-CMIP6 and PMIP3-CMIP5 ensembles show increased temperature seasonality in the Northern Hemisphere resulting from higher obliquity and feedbacks from sea ice and snow cover. These contrasting seasonal responses result in a muted annual-mean temperature changes. Both show

an enhancement of the Northern Hemisphere monsoons and a weakening of the southern hemisphere monsoons. Neither the PMIP4-CMIP6 nor the PMIP3-CMIP5 models show a significant change in the AMOC during the mid-Holocene. This suggests that the changes in wind forcing, temperature gradients, seasonality of sea-ice and precipitation are not sufficient to alter the overall AMOC strength, although investigations into its various components may deliver greater insight.

Although the geographic and seasonal patterns of temperature changes in the PMIP4-CMIP6 ensemble are very similar to those seen in the PMIP3-CMIP5 ensemble, the PMIP4-CMIP6 ensemble is cooler than the PMIP3-CMIP5 ensemble in both summer and winter. This difference is consistent with the change in radiative forcing induced by using realistic GHG concentrations in the PMIP4-CMIP6 (Otto-Bliesner et al., 2017). Advances in the models themselves could also contribute to this difference, for example through their implementation of aerosols. There is a considerable spread in simulated regional *midHolocene* climate between the PMIP4-CMIP6 models. In some cases, for example in the strength of the AMOC, this spread is clearly related to the spread in the *piControl* simulations. Biases in the *piControl* may also help to explain the underestimation of the northward expansion of the NH monsoons, since the global monsoon domain is underestimated by both ensembles in the *piControl* compared to observations.

This preliminary analysis of the PMIP4-CMIP6 *midHolocene* simulations already demonstrates the utility of running palaeo-climate simulations to evaluate the ability of state-of-the-art models to realistically simulate climate change and thus to real-istically simulate the likely trajectory of future climate changes. It showed that relationships between the quality of models representations of the present day and its ability to correctly simulate mid-Holocene climate changes are not straightforward: a finding that holds even for higher resolution models. Although it is disappointing that the PMIP4-CMIP6 simulations are not significantly better than the PMIP3-CMIP5 models in capturing important features of the mid-Holocene climate, analyses of the mechanisms giving rise to these failures should shed light on the need for improved physics and processes in future versions of the CMIP climate models. The examination of how the biases in the *piControl* simulations impact the simulation of past climates is directly relevant to understanding how modern biases are propagated into future projections. Furthermore, the simi-larities between the PMIP4-CMIP6 and PMIP3-CMIP5 simulations provide an argument for combining them to create a single ensemble, which will considerably enhance the statistical power of future analyses. Sensitivity tests, already planned within the framework of PMIP4-CMIP6 (Otto-Bliesner et al., 2017), should help to disentangle the impacts of specific feedbacks on simulated climate changes.

The PMIP4-CMIP6 *midHolocene* simulations provide an opportunity for quantitative evaluation of different aspects of model performance at both global and regional scales. They can be used in process-based analyses to assess the plausibility of future climate change mechanisms (Braconnot and Kageyama, 2015; D'Agostino et al., 2019; Yoshimori and Suzuki, 2019). Palaeoclimate evaluations can then be used to weight models when creating fit-for-purpose ensembles to investigate climate impacts on environmental processes – both in the past and in future projections (Schmidt et al., 2014a). Accurate representation of mid-Holocene climate, through the creation of a best-estimate climate from the PMIP ensembles, would allow us to examine e.g. the role of climate changes on the spread of early agriculture (d'Alpoim Guedes and Bocinsky, 2018; Petraglia et al., 2020). In a similar way, by constraining the choice of future projections to models that can simulate past climate changes well, it would be possible to construct more realistic best-estimates of the impacts of projected climate changes on food security and ecosystem services (Firdaus et al., 2019; Malhi et al., 2020), or on extreme events such as flooding (Boelee et al., 2019).

*Code and data availability.* The necessary output variables from both the *midHolocene* and *piControl* simulations are freely available from the Earth System Grid Federation at https://esgf-node.llnl.gov/search/cmip6/. (HadGEM3-GC31-LL and UofT-CCSM-4 have committed to lodge their data as soon as practical). A GitHub repository is available at https://github.com/chrisbrierley/PMIP4-midHolocene with the code used for this analysis. The Temperature 12k database, along with latitude-zone and global temperature reconstructions using multiple

statistical methods, is available through the World Data Service (NOAA) Paleoclimatology (www.ncdc.noaa.gov/paleo/study/27330). The Bartlein et al. (2011) reconstructions are downloadable as an Electronic Supplementary Material of the article itself. The Compo et al. (2011) Reanalysis can be found at www.esrl.noaa.gov/psd/data/gridded/data.20thC_ReanV2c.html. The precipitation observations of Adler et al. (2003) and Xie and Arkin (1997) are archived at https://www.esrl.noaa.gov/psd/data/gridded/data.cmap.html and https://www.esrl.noaa.gov/psd/data/gridded/data.gpcp.html respectively. The pre-industrial latitudinal average temperatures were created using anomalies of Ilyas et al.

(2017) from https://oasishub.co/dataset/global-monthly-temperature-ensemble-1850-to-2016 combined with the HadCRUT4 (Morice et al., 2012) absolute climatological temperatures from https://crudata.uea.ac.uk/cru/data/temperature/.

*Author contributions.* There are three tiers of authorship for this research, with the latter two in reverse alphabetical order. C.M.B., A.Z., S.P.H. and P.Br. performed the bulk of the writing and analysis. C.J.R.W., D.J.R.T., X.S., J-Y.P., R.O., D.S.K., M.K., J.C.H., M.P.E., J.E-G., R.D'A., D.C., M.C. and P.Ba. contributed text and analysis to the research. The third tier of authors contributed data for the manuscript.

*Competing interests.* The authors declare no competing interests

*Acknowledgements.* We acknowledge the World Climate Research Programme's Working Group on Coupled Modelling, which is responsible for CMIP, and we thank the groups developing the climate models (listed in Tab. 1 & 2 of this paper) for producing and making available their model output. C.M.B., S.P.H., P.Br., C.J.R.W., X.S., R.D'A., M.C. and G.L. received funded by JPI-Belmont Forum project entitled Palaeoclimate Constraints on Monsoon Evolution and Dynamics (PaCMEDy). C.M.B., C.J.R.W. and S.P.H. were funded in part by

NERC (NE/P006752/1). D.J.R.T. and C.M.B. were funded in part by NERC (NE/S009736/1). S.P.H. acknowledges the ERC-funded project GC2.0 (Global Change 2.0: Unlocking the past for a clearer future, grant number 694481). R.O. acknowledges support from the Integrated Research Program for Advancing Climate Models (TOUGOU programme) from the Ministry of Education, Culture, Sports, Science and Technology (MEXT), Japan. The simulations using MIROC models were conducted on the Earth Simulator of JAMSTEC. The NorESM simulations were performed on resources provided by UNINETT Sigma2 – the National Infrastructure for High Performance Computing

and Data Storage in Norway. D.S.K.,C.R. and N.M. were funded by US-NSF-AGS-1602105. P.M. was supported by the state assignment project 0148-2019-0009. E.V. was supported by RSF grant 20-17-00190. B.L.O.-B., E.C.B. and R.T. acknowledge the CESM project, which is supported primarily by the National Science Foundation (NSF). This material is based upon work supported by the National Center for Atmospheric Research (NCAR), which is a major facility sponsored by the NSF under Cooperative Agreement No. 1852977. Computing and data storage resources, including the Cheyenne supercomputer (doi:10.5065/D6RX99HX), were provided by the Computational and

Information Systems Laboratory (CISL) at NCAR. We thank R. Eyles (UCL) for some invaluable database management and pre-processing.

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

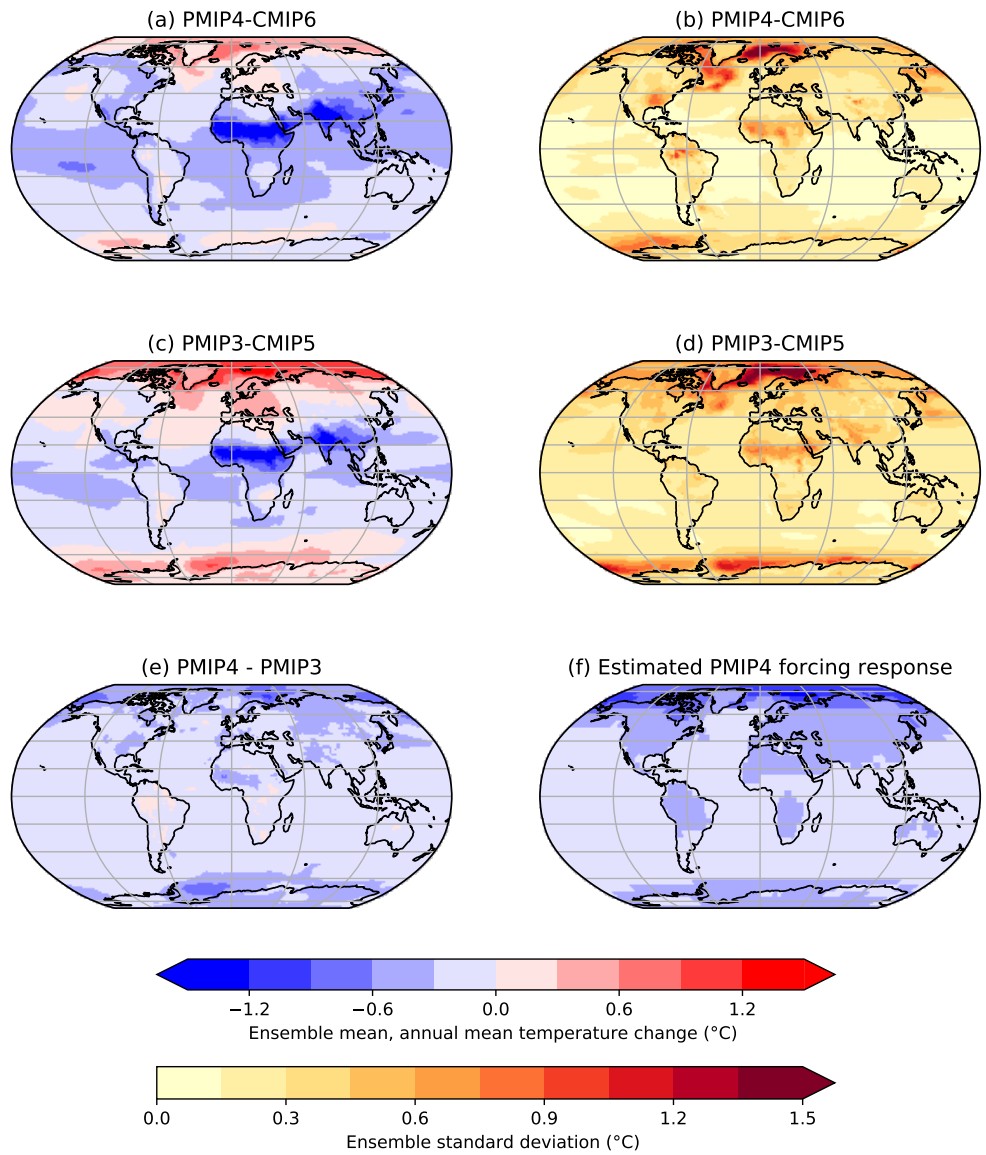

**Figure 1. Annual mean surface temperature change in the *midHolocene* simulations (°C).** (a) The ensemble mean, annual mean temperature changes in PMIP4-CMIP6 (*midHolocene - piControl*) and (b) the intermodel spread (defined as the across ensemble standard deviation). (c) The ensemble mean, annual mean temperature change in PMIP3-CMIP5 and (d) its standard deviation. (e) The difference in temperature between the two ensembles. (f) The estimated response to the greenhouse gas concentration reductions in the experimental protocol.

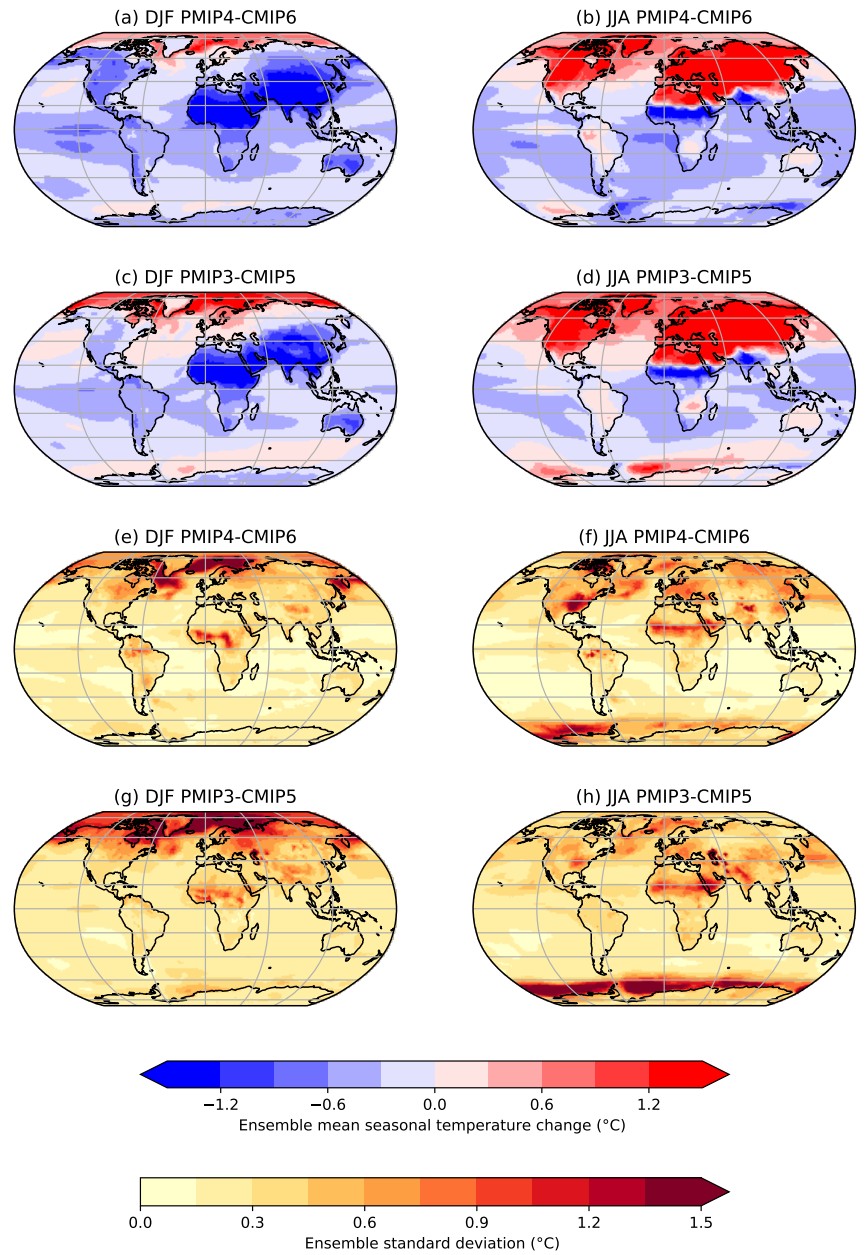

**Figure 2. Seasonal surface temperature changes in the *midHolocene* simulations** (°C). (a,b) The ensemble mean temperature changes in PMIP4-CMIP6 (*midHolocene - piControl*) in DJF and JJA. (c,d) The ensemble mean temperature changes in PMIP3-CMIP5 in DJF and JJA. The intermodel spread (defined as the across ensemble standard deviation) in seasonal temperature changes seen across the ensembles: (e) DJF in PMIP4-CMIP6, (f) JJA in PMIP4-CMIP6, (g) DJF in PMIP3-CMIP5 and (h) JJA in PMIP3-CMIP6.

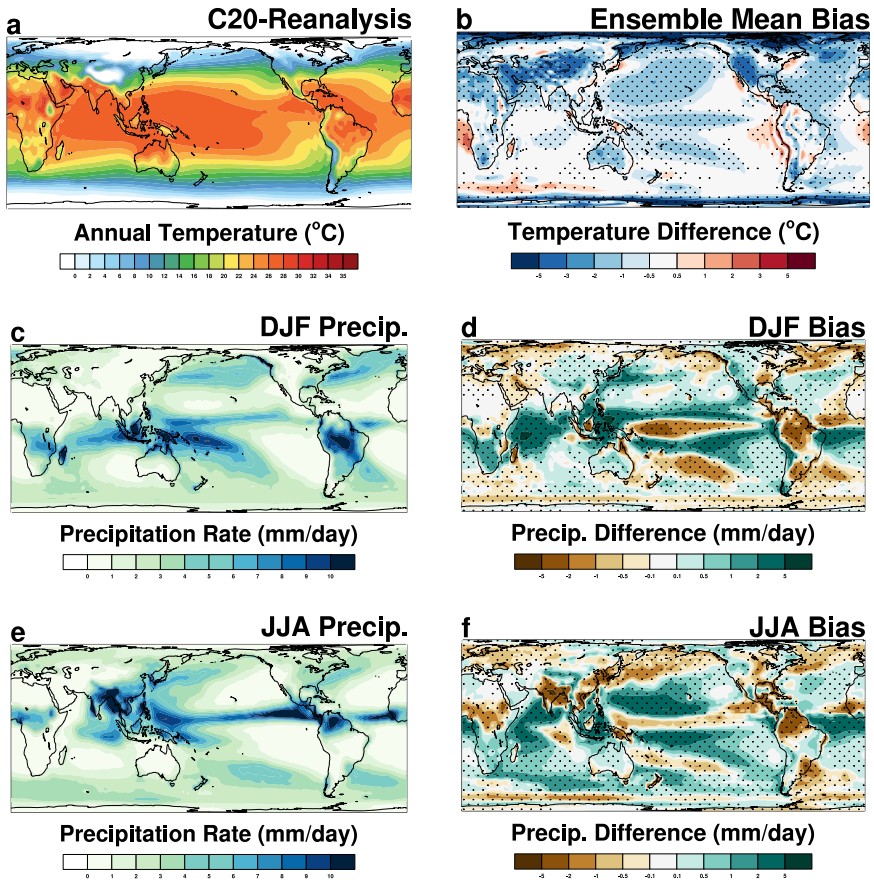

**Figure 3. Comparison of the CMIP6 ensemble to observations**. (a) The annual mean surface temperatures in the C20 Reanalysis (Compo et al., 2011) between 1881-1900. (b) The ensemble mean difference in annual surface air temperature from the C20 Reanalysis within the *piControl* simulations. Ability of the ensemble to simulate the seasonal cycle of precipitation for the present-day. (c,e) The precipitation climatology seen in the GPCP (Adler et al., 2003) observational dataset between 1971-2000 for DJF and JJA respectively. (d,f) The ensemble mean difference in seasonal precipitation from GPCP within the *piControl* simulations for DJF and JJA respectively. Stippling indicates that two-thirds of the models agree on the sign of the bias.

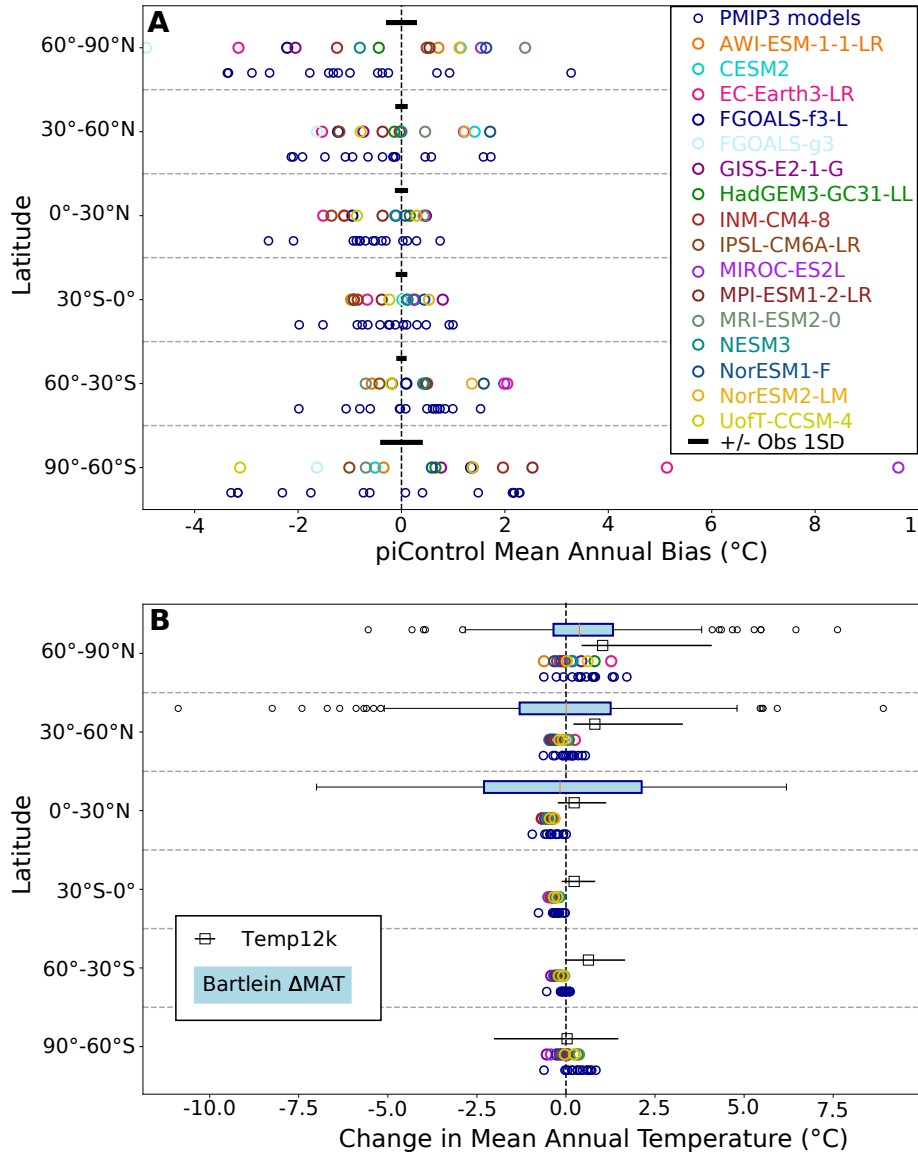

**Figure 4. Zonal averaged temperatures within the PMIP4-CMIP6 ensemble**. (a) Comparison of the *piControl* zonal mean temperature profile of individual climate models to the 1850-1900 observations. The area-averaged, annual mean surface air temperature for 30°latitude bands in the CMIP6 models (identified), CMIP5 models (blue circles) and a spatially complete compilation of instrumental observations over 1850-1900 (black, Ilyas et al., 2017; Morice et al., 2012). (b) The simulated annual mean temperature change averaged over 30° zonal bands for each of the individual CMIP6 models. The equivalent changes estimated from the Temperature 12k compilation (Kaufman et al., 2020b) via a multi-method approach are shown along with their 80% confidence interval. The distribution of Bartlein et al. (2011) reconstructed temperatures within each latitude bands are shown in the NH, because the tropical and southern hemisphere latitudes are only represented by sites in Africa. *The data points for all models, as well as the equivalents over land or ocean, are provided in Table S3.*

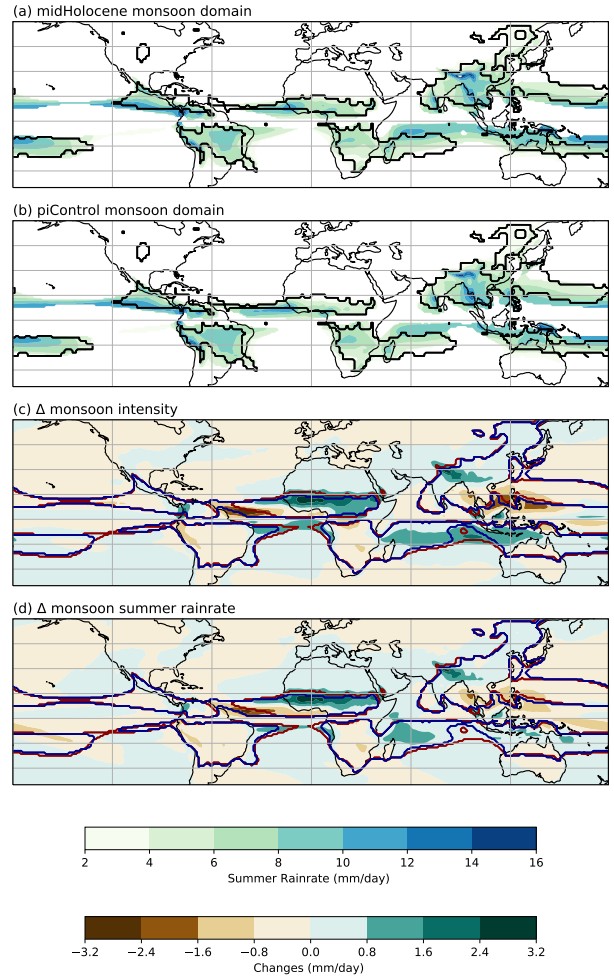

**Figure 5. PMIP4-CMIP6 ensemble mean global monsoon domain (mm/day)**. The monsoon domain for each simulation is identified by applying the definitions of Wang et al. (2011) and in sect. 2.2 to the PMIP4-CMIP6 ensemble mean of both (a) the *midHolocene* and (b) the *piControl* simulations. The black contour in (a,b) shows the boundary of the domain derived from present-day observations (Adler et al., 2003). The simulated changes in the monsoon domain are determined by changes in both (c) the monsoon intensity – average rain rate difference between summer and winter – and (d) the summer rain rate. In (c,d) the red and blue contours show the boundary of *midHolocene* and *piControl* global monsoon domains respectively.

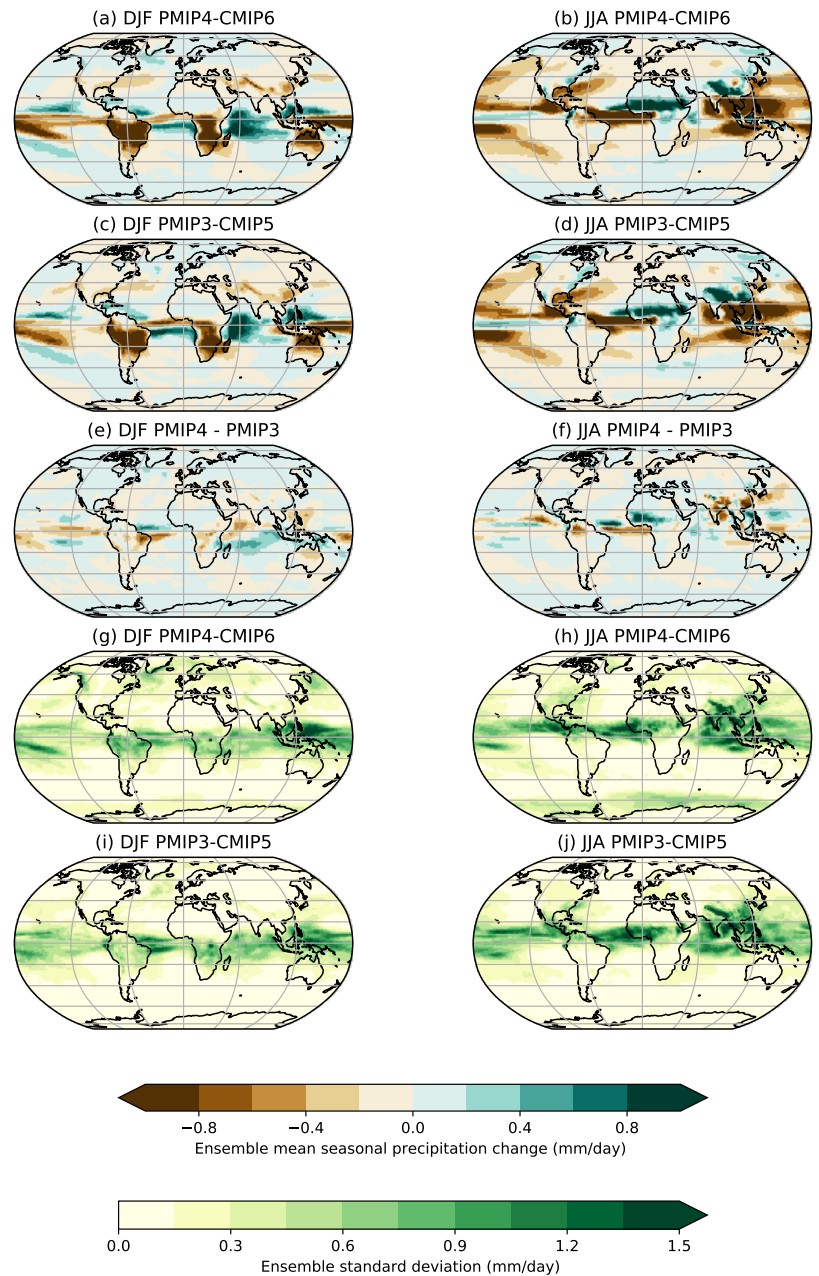

**Figure 6.** *midHolocene* **seasonal changes in precipitation (mm/day).** (a,b) The ensemble mean precipitation changes in PMIP4-CMIP6 (*midHolocene - piControl*) in DJF and JJA. (c,d) The ensemble mean precipitation changes in PMIP3-CMIP5 in DJF and JJA. (e,f) The differences in DJF and JJA precipitation between the PMIP4-CMIP6 and PMIP3-CMIP5 ensembles. The intermodel spread (defined as the across ensemble standard deviation) in seasonal precipitation changes seen across the ensembles: (g) DJF in PMIP4-CMIP6, (h) JJA in PMIP4-CMIP6, (i) DJF in PMIP3-CMIP5 and (j) JJA in PMIP3-CMIP6.

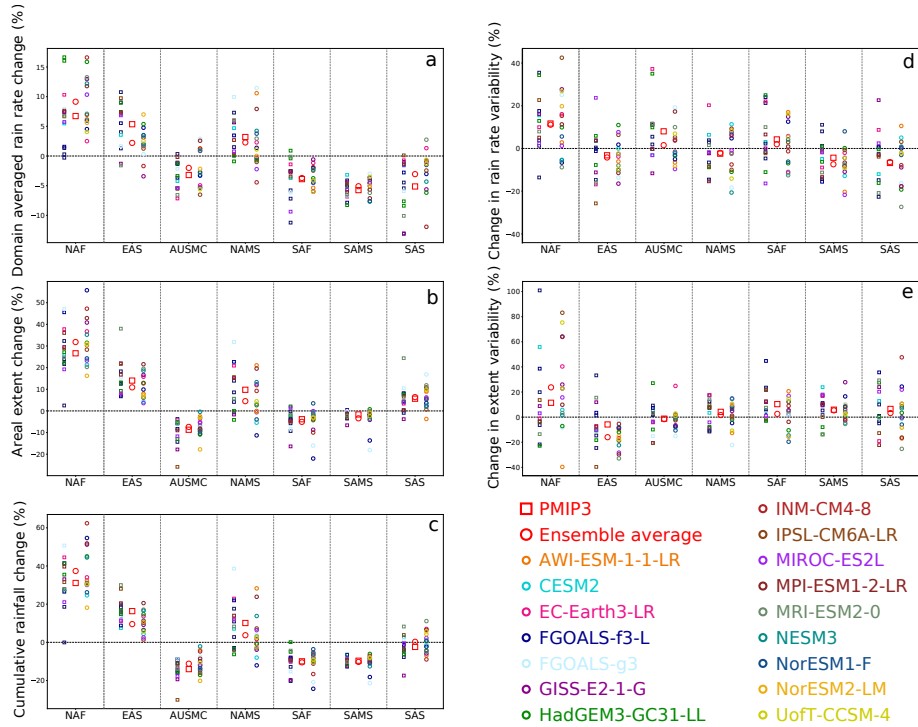

**Figure 7. Relative changes in individual *midHolocene* monsoons.** Five different monsoon diagnostics (see sect. 2.2) are computed for each of seven different regional domains (Christensen et al., 2013). (a) The change in area-averaged precipitation rate during the monsoon season (MJJAS) for each individual monsoon system. (b) The change in the areal extent of the regional monsoon domains. (c) The percentage change in the total amount of water precipitated in each monsoon season (computed as the precipitation rate multiplied by the areal extent). (d) Change in the standard deviation of interannual variability in the area-averaged precipitation rate. (e) The change in standard deviation of the year-to-year variations in the areal extent of the regional monsoon domain. The abbreviations used to identify each regional domain are: North America Monsoon System (NAMS), North Africa (NAF), Southern Asia (SAS) and East Asia summer (EAS) in the Northern Hemisphere and South America Monsoon System (SAMS), South Africa (SAF) and Australian-Maritime Continent (AUSMC) in the Southern Hemisphere.

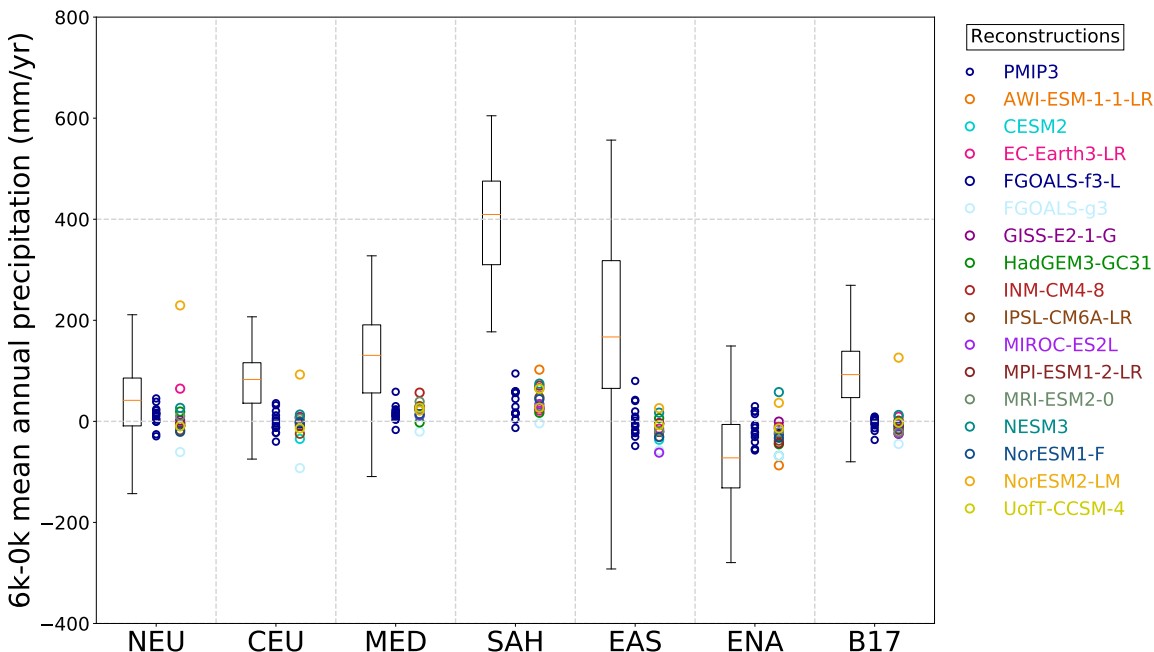

**Figure 8. Comparison between simulated annual precipitation changes and pollen-based reconstructions** (from Bartlein et al., 2011). Seven regions where multiple quantitative reconstructions exist are chosen. Six of them are defined after Christensen et al. (2013), and are Northern Europe (NEU), Central Europe (CEU), the Mediterranean (MED), the Sahara/Sahel (SAH), East Asia (EAS) and Eastern North America (ENA). Mid-continental Eurasia (B17) is specified by Bartlein et al. (2017) as 40–60°N, 30-120°E. The distribution of reconstructions within the region are shown by boxes and whiskers. The area-averaged change in mean annual precipitation simulated by CMIP6 (individually identifiable) and CMIP5 (blue) within each region is shown for comparison. (After Flato et al., 2013)

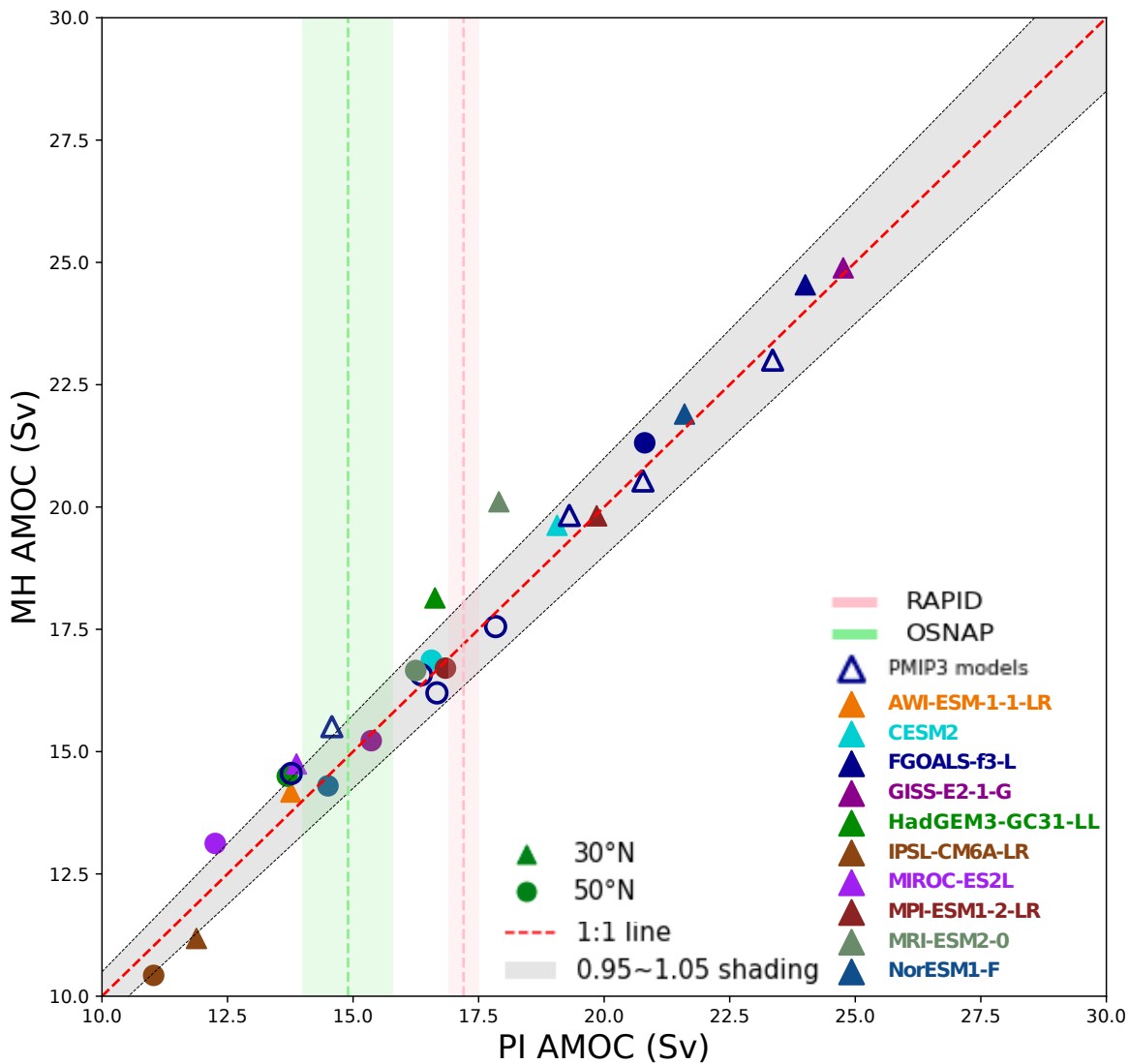

**Figure 9. Atlantic Meridional Overturning Circulation in the simulations**. The strength of the AMOC is defined as the maximum of the mean meridional mass overturning streamfunction below 500m at 30°and 50°N in the Atlantic. The strength in the *piControl* simulation provides the horizontal axis, whilst the vertical location is given by the strength in the *midHolocene* simulation. Data points lying on the 1:1 line demonstrate no change between the two simulations. Observational estimates of the present-day AMOC strength are shown from both the RAPID-MOCHA array (at 26°N, Smeed et al., 2019) and the OSNAP section (between 53°N and 60°N, Lozier et al., 2019).

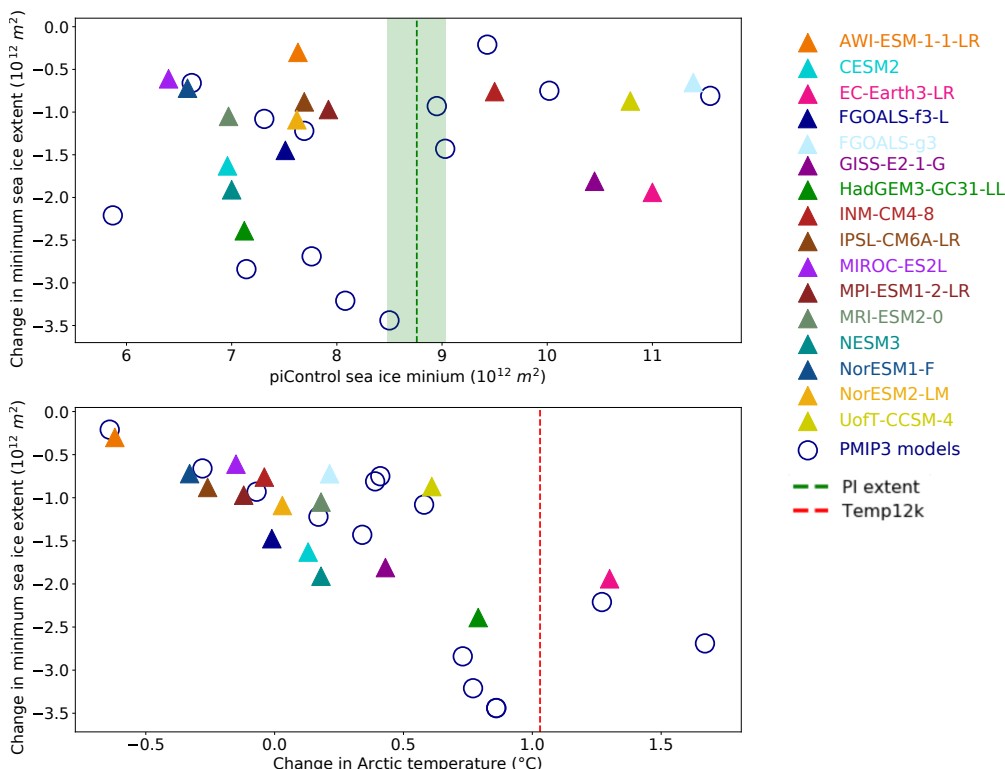

**Figure 10. Changes in Arctic sea ice minimum extent**. The change in the areal extent of the minimum sea ice cover (i.e. gridboxes with greater than 15% concentration) at the mid-Holocene compared to (a) the minimum sea ice extent in the piControl simulations and (b) the Arctic annual mean temperature change. Observational estimates of the pre-industrial extent (Walsh et al., 2016) and mid-Holocene Arctic warming (Fig. 4; Kaufman et al., 2020a) are also shown.

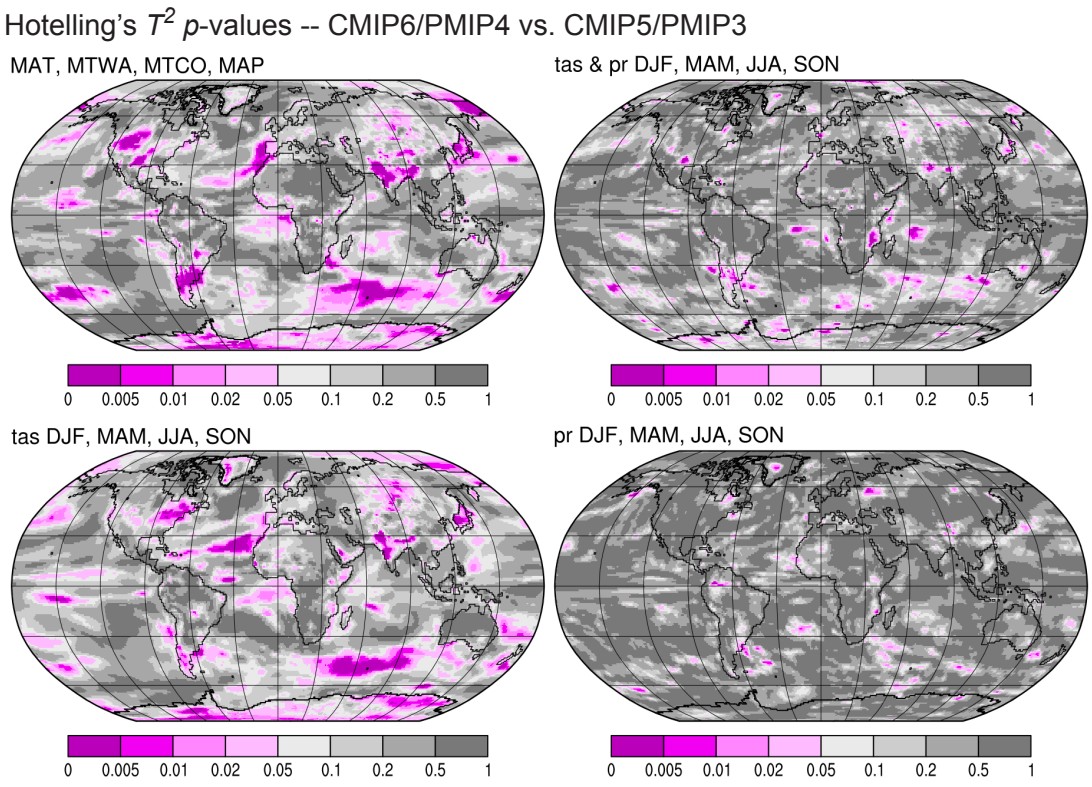

**Figure 11. Maps of the p-values of Hotelling's** $\mathrm{T}^2$ **test** (Wilks, 2011) comparing the PMIP4-CMIP6 and PMIP3-CMIP5 ensembles. Four different combinations of the key variables analysed here are assessed (given in the top left above the panels). Values less than 0.05 would ordinarily be considered to be significant, but the total number of such values on each individual map does not exceed the false discovery rate. Harrison et al. (2015) presents equivalent analysis comparing PMIP3-CMIP5 with PMIP2-CMIP3 (using the variables in the top left panel).

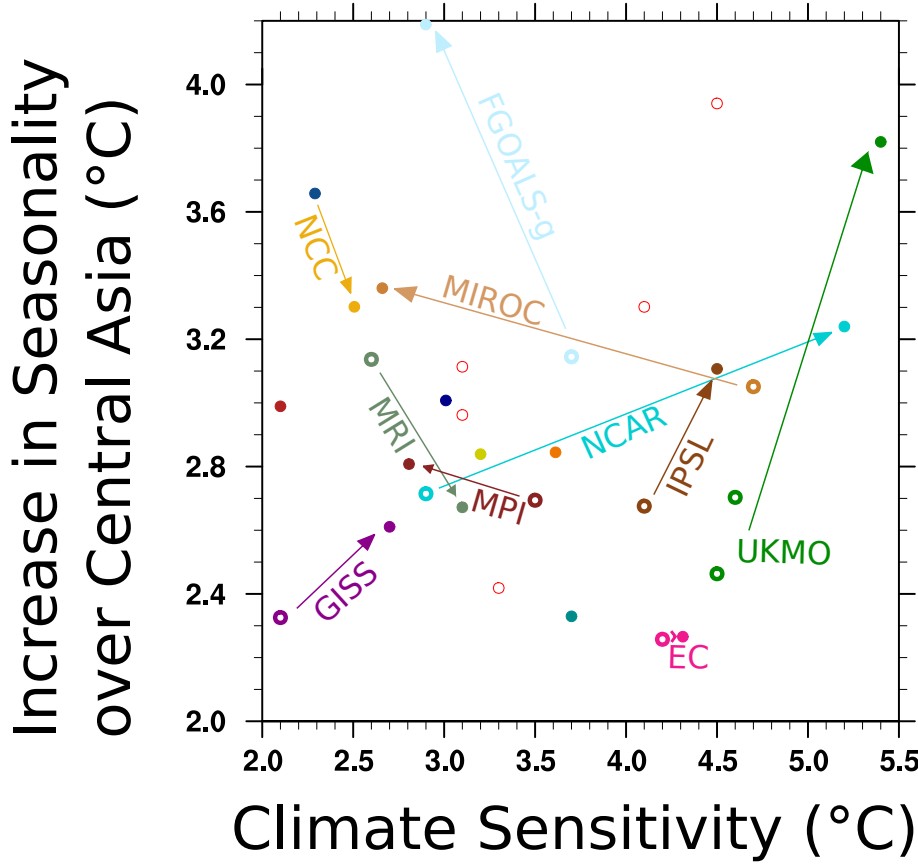

**Figure 12. The relationship between equilibrium climate sensitivity and increasing seasonality over Central Asia**. The seasonality is computed as the mean temperature of the warmest month minus the mean temperature of the coldest month, averaged over 30–50°N, 60–75°E (Christensen et al., 2013). The shifts between different generations of models are indicated, and labelled after their modelling group (NCAR developed both CCSM4 & CESM2; NCC developed NorESM1-F & NorESM2-LM; UKMO developed HadGEM2-CC, HadGEM2-ES & HadGEM3-GC31-LL).

**Table 1.** Models contributing *midHolocene* simulations under CMIP6. See Table S1 for further information about the individual simulations.

| model | $\Delta T^{eq}_{2xCO_2}$ (K) | *midHolocene* length† (yrs) | *piControl* length† (yrs) | Model Reference | Expt Ref. & Notes |
|---|---|---|---|---|---|
| AWI-ESM-1-1-LR | 3.6 | 100 | 100 | Sidorenko et al. (2015) | Dynamic Vegetation |
| CESM2 | 5.3 | 700 | 1200 | Gettelman et al. (2019) | Otto-Bliesner et al. (2020a) |
| EC-Earth3-LR | 4.3 | 200 | 200 | Wyser et al. (2019) | – |
| FGOALS-f3-L | 3.0 | 500 | 561 | Wang et al. (2020) | – |
| FGOALS-g3 | 2.9 | 500 | 200 | He et al. (2020) | – |
| GISS-E2-1-G | 2.7 | 100 | 851 | Bauer and Tsigardis (2020) | – |
| HadGEM3-GC31-LL | 5.4 | 100 | 100 | Williams et al. (2018) | Williams et al. (2020) |
| INM-CM4-8 | 2.1 | 200 | 531 | Volodin et al. (2018) | – |
| IPSL-CM6A-LR | 4.5 | 550 | 1200 | Boucher, et al. (2020) | TSI of 1361.2 $W/m^2$ |
| MIROC-ES2L | 2.7 | 100 | 500 | Hajima et al. (2020) | Ohgaito et al. (2020) |
| MPI-ESM1-2-LR | 2.8 | 500 | 1000 | Mauritsen et al. (2019) | – |
| MRI-ESM2 | 3.1 | 200 | 701 | Yukimoto et al. (2019) | – |
| NESM3 | 3.7 | 100 | 100 | Cao et al. (2018) | – |
| NorESM1-F | 2.3 | 200 | 200 | Guo et al. (2019) | – |
| NorESM2-LM | 2.5 | 200 | 200 | Seland et al. (2020) | – |
| UofT-CCSM-4 | 3.2 | 100 | 100 | Chandan and Peltier (2017) | TSI of 1360.89 $W/m^2$ |

†The lengths given are the number of simulated years used here to compute the diagnostics. These years are taken after the model has been spun-up.

**Table 2.** Models contributing *midHolocene* simulations under CMIP5. See Table S1 for links to each individual simulation.

| model | $\Delta T_{2xCO_2}^{eq}$ (K) | *midHolocene* length† (yrs) | *piControl* length† (yrs) | Reference |
|---|---|---|---|---|
| bcc-csm1-1 | 3.1 | 100 | 500 | Xin et al. (2013) |
| CCSM4 | 2.9 | 301 | 1051 | Gent et al. (2011) |
| CNRM-CM5 | 3.3 | 200 | 850 | Voldoire et al. (2013) |
| CSIRO-MK3-6-0 | 4.1 | 100 | 500 | Jeffrey et al. (2013) |
| CSIRO-MK3L-1-2 | 3.1 | 500 | 1000 | Phipps et al. (2012) |
| EC-Earth-2-2 | 4.2 | 40 | 40 | Hazeleger et al. (2012) |
| FGOALS-G2 | 3.7 | 680 | 700 | Li et al. (2013) |
| FGOALS-S2 | 4.5 | 100 | 501 | Bao et al. (2013) |
| GISS-E2-R | 2.1 | 100 | 500 | Schmidt et al. (2014b) |
| HadGEM2-CC | 4.5 | 35 | 240 | Collins et al. (2011) |
| HadGEM2-ES | 4.6 | 101 | 336 | Collins et al. (2011) |
| IPSL-CM5A-LR | 4.1 | 500 | 1000 | Dufresne et al. (2013) |
| MIROC-ESM | 4.7 | 100 | 630 | Sueyoshi et al. (2013) |
| MPI-ESM-P | 3.5 | 100 | 1156 | Giorgetta et al. (2013) |
| MRI-CGCM3 | 2.6 | 100 | 500 | Yukimoto et al. (2012) |

†The lengths given are the number of simulated years used here to compute the diagnostics. These years are taken afer the model has been spun-up.