# Peer review of "Large-scale features and evaluation of the PMIP4-CMIP6 *midHolocene* simulations"

_Climate of the Past, 2019_

## Referee Comment (RC1) · Anonymous Referee #1 · 30 Jan 2020

In this manuscript, the authors have evaluated the PMIP4-CMIP6 simulations along with PMIP3-CMIP5 simulations. They found that there is no significant difference in the simulated climate between these two sets of simulations. The manuscript is well written and I would like to suggest this manuscript to be accepted with some minor revision.

Comments: 1. page 2, line 8, change "cooler that" to "cooler than". 2. page 3, figure 3 caption, is 1981-1900 correct? or it should be 1900-1981 or some other time periods. 3. For figure 2, it might be a good idea to add two panels to represent the difference between PMIP4 and PMIP3 as shown in Figure 6. Also, it seems that the standard deviation is not discussed in any detail in the manuscript. If so, those panels could be removed from Figures 2 and 6. 4. It might be a good idea to test the significance for

Figure 6e,f in stead of just use Figure 10 to show the multi-variables test. 5. For the standard deviation in Figures 2 and 6, it is not clear whether it is calculated from each individual model first, then averaged across models or it is the across model standard deviation.

---

## Author Comment (AC1) · 30 Jan 2020

We would like to thank Reviewer 1 for their speedy and supportive comments on our manuscript.

I would like to clarify that the standard deviations presented in figures 1,2 & 6 are intended to demonstrate the variation in the signal between the GCMs. They are the ensemble standard deviation in the temporal mean changes at each location - rather than the ensemble means of the temporal standard deviations.

We would correct the spelling and captioning errors in a revised manuscript. We would consider the proposed changes to the figures.

---

## Referee Comment (RC2) · Anonymous Referee #2 · 11 Feb 2020

Review of manuscript entitled "Large-scale features and evaluation of the PIM4-CMIP6 midHolocene simulations, by Brierley et al.

This manuscript presents the very recent simulations performed within the PIM4-CMIP6 project and aiming at reproducing the climate of the mi-Holocene (6000 years ago). This climate was characterized by specific external forcing (greenhouse gases concentrations) and orbital parameters, which have been prescribed in a consistent way in the different models, following a previous study (Otto-Bliesner et al. 2017). At the time of the publication, twelve models have performed these simulations and are analyzed here, which is equivalent to the number of models having performed the previous coordinated exercise. The manuscript proposes a first analysis of four specific features of the midHolocene climate: surface temperature, monsoons, extratropical

hydrological responses and the ocean circulation. A final section discusses general assessments of the midHolocene climate.

The paper is in general very well written and presented, and figures are of high quality. References are thoroughly cited. This paper is clearly an important milestone for the PIMP4-CMIP6 intercomparison process, benchmarking the publication of the simulations and probably calling for subsequent more original analysis. I only have a few major comments and a series of minor ones listed below. I want to indicate to the authors that I am not a paleo-expert, I come from the climate modeling community, typically rather focusing and the historical and future periods.

Main comments:

1. On the calendar adjustment( section 2.2)

- I am surprised that this calendar adjustment has to be performed offline. Is there a fundamental reason that the protocol does not include a change of the online calendar?

- Could you provide a comparison of the effect of the calendar adjustment for a variable for which it is proven useful (temperature?). Here, you only give numbers for the monsoon system, which you claim is not suited for this adjustment.

- It sounds illogical to me to assess (in this case to reject) the reliability of the calendar adjustment for the monsoon system using a fixed domain (lines 95-98) and just after (lines 99 and following) explain that the spatial extent of the monsoon varies so that the latter has to be adjusted for future assessments. Could you justify or modify?

2. Section 3.5 is relatively confusing.

- L. 304-307: I have to say that I don't understand the point that is made here. First, grammatically, I don't understand the "they" l. 305. And logically, I don't understand the link of what is said here with the beginning of the paragraph.

- L. 309 "noisy" would be more accurate than "chaotic" I think.

- Paragraph beginning l. 320: I find this paragraph difficult to follow and perhaps not so well organized:

o L. 322-323: it is not clear to me how Fig. 1 shows that the protocol is responsible for the detected differences.

o The fact that "there is no inherent relationship between climate sensitivity and seasonality is repeated twice (l. 324 and l. 326) with no obvious demonstration in-between.

o It could/should be said more explicitly that Fig. 11 is just an example and that checks have been performed for other features of the miHolocene climate features (have they?)

o I don't fully understand the last sentence of the paragraph: in what sense could it provide a constrain then?

Minor comments

- l. 129: I don't know what recrine (212) is, please explain.

- lines 35-38: the increased climate sensitivity in the CMIP6 generation of climate models is still under investigation and I think it is worth specifying: add "The reasons for this increase is still under investigation, but it may influence also the sensitivity of the models to the midHolocene external forcing.

- End of section 2.1: on the specificities of the protocol and the models differences, I would like to advertise the site https://es-doc.org/ that provides a detailed documentation of all these points, with a full documentation of each model.

- L. 138: as "as compared to piControl conditions" after "Europe" and specify Fig. 1a.

- L. 155-156: From Fig. 1, I am not completely sure that I can state that "the change in MAT with respect to the piControl in the PIMP4-CMIP6 ensemble is less than in the PIM3-CMIP5: I see indeed a weaker warming at high latitudes but a stronger cooling in the tropics. Please correct or clarify.

[Figure]

- L. 166: I would suggest naming explicitly the C20 reanalysis in the brackets.

- L. 170-171: as shown in Fig. 1(e), the PIMP4-CMIP6 models are generally colder than the PIMP3-CMIP5 ones. The reduced warm bias along the EBUS could simply be a consequence of this. Thorough analysis of the evolution of this bias would require the use of relative temperature (e.g. Hourdin et al. 2015, GRL)

- L. 175: I don't understand how this correlation coefficient is computed.

- L. 190: I would rather use the term "intermodel spread", as in several other places of the manuscript. Standard deviation typically relates to temporality in my view. Changes in the legend of Fig. 1, 2 and 6 are also needed.

- L. 233-235: the term "the changes of precipitation" is sometimes confusing: I think you speak of the changes of each PIMP exercise with respect to piControl, but one could also read as the changes from PIMP3 to PMIP4. I suggest clarifying.

- L. 239: remove the word "indeed"

- L. 250 this sentence does not mean much to me. Large spread with respect to what?

- L. 277 I would rather write " a decline of up to ∼15%"

- L. 280: It should be made clear that the magnitude of the AMOC computed at 50°N in z-coordinates is a little bit misleading, since in fact intense transport of water mass occurs at similar depths, but with very different densities (e.g. Zhang GRL 2010).

- L. 286: I don't clearly understand the link that is made with the magnitude of low frequency internal variability in AMOC. Clarify.

- Conclusion section will have to be changed accordingly to previous remarks (on section 3.5 in particular

- L. 365: I have no proof at this stage that the changes in the implementation of aerosols in CMIP6 is an improvement.

---

## Referee Comment (RC3) · Anonymous Referee #3 · 17 Feb 2020

The manuscript presents the recent simulations of mi-Holocene performed within the PIM4-CMIP6. It is a valuable and interesting work as it will be one of the main references for the future studies. The text is written and structured well, and the storyline is what would be expected from such a paper but I have few main critical and some major/minor comments as are listed below. Overall, I think it should be published in Climate of the Past, after a minor-moderate revision.

Main comments: As one of the important components of the climate, I wonder why the sea ice story was not included except the short sentence at line 193. I suggest to add the 2-D maps of sea ice concentration in the Arctic (for both summer and winter) which would also be relevant to the AMOC story. For instance, when there is sea ice covering part of the deep water formation (DWF) region in the Labrador Sea (due to model bias),

DWF is reduced. If the model has too much sea ice over the Labrador Sea in both PI and midHolocene, then that can partially explain why the AMOC does not differ much between the two periods, as any freshening or cooling cannot influence the DWF.

With regards to AMOC, it would be nice to say something (few sentences) about the regions where deep water formation happens based on mixed layer depth values in March. It could be that if there is sea ice over the Labrador (recalling my previous comment), then the region of deep water formation might shift. . .

Again about AMOC, I know that normally 30°N (models) or 26.5°N (observations) are chosen for calculating the maximum AMOC value. Since this latitude of maximum AMOC can differ between models as well as the two periods of study, I would not pick one latitudinal point. Instead, I will define a range e.g., 25-35°N to calculate the maximum AMOC. I also suggest to add the observations to your plot (RAPID-MOCHA array observations Smeed et al., 2017 http://www.rapid.ac.uk/rapidmoc/rapid_data/transports.php).

My major and minor comments regarding the text and analysis are following:

Methods: The mean values, are they averaged over the entire simulated years mentioned in Table 1?

Line 86: Can you say in one sentence how is PaleoCalAdjust performing in general?

Lines 88-98: For the annual mean I understand you do not need calendar adjustment. But if you use your daily values from PaleoCalAdjust and make the annual mean, how much would it differ from the main annual mean? This can give you some ideas about the potential interpolation errors (if there is no original daily data).

Line 98: I do not understand "we have therefore..." so you use the method when you thin it is good?

Line 107: in this line and any other lines (line 219) please change "interannual variability" to → internal climate variability because the variability is not only interannual. . .

Lines 154-161: move these lines to after line 140.

Line 171: "...colder conditions over the Labrador current..." which figure you are referring to? And I assume you meant Labrador Sea and not current?

Line 174: not only in the tropics but over the oceans in general there is a better match

Line 234: ". . . change in precipitation" change between what?

Line 234-236: you used "change" three times in one sentence, modify please and combine it with the previous sentence.

Line 324: role

Line 374: ". . . need for improved physics and processes..."

Figure 3 caption: check the years where observation was used e.g., you wrote "1981-1900". . . also check the rest of caption.

Figure 5: would it be possible to make the similar figure for the observation/reanalysis?

Please also note the supplement to this comment:
https://www.clim-past-discuss.net/cp-2019-168/cp-2019-168-RC3-supplement.pdf

———————————————————

---

## Short Comment (SC1) · 5 Mar 2020

Figure 3, as one of the reviewers pointed out, the caption for Fig 3 states a date range of 1981-1900 for the C20 Reanalysis. The main text indicates this should be 1871-1900 (Line 166).

Lines 316-317, I believe the second part of this sentence is referring to PMIP2 (Harrison et al., 2015), not PMIP3.

Line 323: protol -> protocol

---

## Author Comment (AC2) · 1 Apr 2020

Dear Dr Alder,

Thank you for your comments. You're correct about the figure caption being incorrect - it is supposed to say this is the early industrial period of 1871-1900. I hadn't spotted this issue about the PMIP generation, but you're completely right and it should be PMIP2-CMIP3.

Cheers, Chris
* * *

---

## Author Comment (AC3) · 8 Apr 2020

We would like to thank you for your diligent review. We believe that we can address the concerns you have raised in a revised manuscript – and that in doing so, the manuscript will be more helpful to a broader audience.

It is obvious that our discussion of the calendar issues has been unclear (it was raised by Reviewer 3 as well). We shall rephrase this subsection in the revised manuscript to improve its clarity. The insolation changes resulting from the altered orbital configuration are a key part of the experiment protocol. The problem that the adjustment resolves is to do with the aggregating of data during run-time to create monthly-resolution output. To fix this online can require substantial alteration of a model's output processing

code, which would act as hurdle to participation in PMIP. The calendar adjustment has never previously been implemented in a multi-model study, despite several calls for it (e.g. Kutzbach & Gallimore 1988; Joussaume & Braconnot, 1997). The creation of easy-to-use software by Bartlein & Shafer (2019) has meant it has been practical to include it for the first time here. This review requests a justification and assessment of its impact on surface temperature and precipitation. These are demonstrated in detail within Bartlein & Shafer (2019), but we shall a summary in the revised manuscript. We shall also correct the apparent contradiction raised by the juxtaposition of the discussion of fixed and varying monsoon domains during revision.

The review notes that Section 3.5 (on comparing the PMIP4 models to data) is relatively confusing, and then provides some specific sentences and paragraphs that were unclear or ambiguous. We shall rewrite this whole section to improve its clarity – in part by taking things more slowly. Finally, the review identifies a series of minor comments about specific sentences or words. We shall address each of these individually.

Given the unfortunate rush to submit papers before 2020, we had knowingly not adopted best practice in CMIP6 data citation and documentation. It was always our intention to have completed this at revision stage (whilst we incorporate the additional simulations that have since been uploaded to the ESFG), and we appreciate the timely reminder.

---

## Author Comment (AC4) · 8 Apr 2020

We would like to thank you for your kind, yet thorough, comments about our manuscript. We believe that the revisions we plan to implement should satisfactorily address your comments.

The first main comment in this review related to our decision not to present findings about sea ice cover changes. There are two main reasons that we did not include sea ice in this manuscript. Firstly, we wanted to constrain the scope of this manuscript to a manageable amount of analyses. It already feels possibly too long with its current 11 figures. The mid-Holocene sea ice story can support a whole manuscript on its own, as demonstrated by the submission of a paper on the lig127k sea ice to this special

issue (Kageyama et al., 2020). Secondly, there are technical issues around the calendar adjustment using the Bartlein and Shafer (2019) software. It has been tested and evaluated on surface temperature and precipitation. The software development required for it to adjust fields on rotated grids has been completed, but it has not yet been scientifically validated for sea ice coverage. Three of the models have, however, provided daily fields that avoid such issues. As a supplement to this Author Comment, we include figures of the composited patterns at the day of the annual maximum and minimum of the Arctic sea ice coverage for this small subset and our preliminary analysis of the role of calendar adjustment on one of them. Given the two reasons it may be best to leave the presentation of sea ice coverage changes at the mid-Holocene for a subsequent more-detailed manuscript (as happened in PMIP3). Conversely, the Last Interglacial equivalent to this manuscript (Otto-Bliesner et al., cp-2019-174) already includes mid-Holocene sea ice coverage changes in its Fig 17. We will explore both approaches before commiting to a course, and take guidance from the Editor upon this question.

The second main comment discusses the AMOC. A sub-group of our authors have already initiated a detailed analysis of the mid-Holocene AMOC and deep-water formation, but this is planned as a separate paper to allow the analysis to sufficiently investigate the mechanisms at play. The review also suggests a subtly different choice to the AMOC latitude. We have tested the alternate definition on a subset of the models (see supplement) and find that it results in variations in the midHolocene AMOC percentage change with a magnitude of at most 0.85

We shall detail retrospectively how we have revised the manuscript in light of each specific comment in the review. However, we feel it would be instructive to respond to two particular ones at this early stage.

Firstly, it is possible to create a version of Fig. 5 that includes the observations/reanalysis, and we include it in the supplement to this Author Comment. However, given that the boundary of the domain in the observations/reanalysis is already

marked in two panels, we question how much more information such an additional panel conveys.

Secondly, the review posits an interesting method to estimate the size of the interpolation error from the PaleoCalAdjust routine – namely by looking at the changes in the annual mean surface temperatures. We include also this figure in the supplement to this Author Comment, and it is up to 0.8oC in magnitude. However, it is important to stress that this difference really arises from assumptions in the subsequent workflow, rather than the PaleoCalAdjust routine itself. We have relied upon the Climate Variability Diagnostics Package (Phillips et al, 2013) to compute the bulk of the fields presented in the manuscript. Within this package, the annual values are calculated as the unweighted average of the 12 monthly values. This results in minimal errors under present orbital configuration yet allows the package to readily handle many different calendars efficiently. Unfortunately, such an assumption is not appropriate once PaleoCalAdjust has been implemented on the Mid-Holocene monthly output, because that software intentionally adjusts the data to represent the different, non-equal length months.

Please also note the supplement to this comment:
https://www.clim-past-discuss.net/cp-2019-168/cp-2019-168-AC4-supplement.zip

---

## Author Response (AR1)

**Reply to the reviewers' comments: Large-scale features and evaluation of the PMIP4-CMIP6 midHolocene simulations** (cp-2019-168)**

C. M. Brierley *et al.* **Correspondence:** c.brierley@ucl.ac.uk

**Summary of changes**

We have adopted many of the revisions to the text suggested to the reviewers. Two of the reviewers commented on the use of the PaleoCalAdjust software, and we shall rewrite that sub-section to provide better explanation and justification. We intend to work further on the text surrounding the data-model analysis in the light of Reviewer 2's comments (Section 3.5). We shall

5 make the improvements to the figures as requested. Further simulations have since become available from at least 3 models and we plan to incorporate them into our analysis. Finally, We have devoted more effort to applying best practice for CMIP6 data citation and documentation within the manuscript.

Blue text below is our response to the reviewer's comments (reproduced in black).

**10 Reviewer 1**

In this manuscript, the authors have evaluated the PMIP4-CMIP6 simulations along with PMIP3-CMIP5 simulations. They found that there is no significant difference in the simulated climate between these two sets of simulations. The manuscript is well written and I would like to suggest this manuscript to be accepted with some minor revision.

We would like to thank the reviewer for their kind comments and are happy to make the revisions suggested

**15**

page 2, line 8, change 'cooler that' to 'cooler than'. Done

page 3, figure 3 caption, is 1981-1900 correct? or it should be 1900-1981 or some other time periods.20 This should have been 1871-1900, and has now been corrected.

For figure 2, it might be a good idea to add two panels to represent the difference between PMIP4 and PMIP3 as shown in Figure 6.

We did initially include these panels during the paper drafts. However, the ensemble differences are pretty consistent in both

25 seasons - and they both look similar to Fig 1e. We now explicitly state "The pattern of cooling in both seasons is very similar to the annual mean ensemble difference in Fig. 1e (not shown)"

It might be a good idea to test the significance for Figure 6e,f instead of just use Figure 10 to show the multi-variables test. Given that the more-sophisticated multi-variate analysis demonstrates that the PMIP3 & PMIP4 ensembles can be treated as

30 a single super-ensemble, we feel such analysis risks giving a false positive. However, we now shout forward to the formal significance testing at the relevant location: "(testing the significance of the differences between the ensembles is discussed in sec. 3.5)"

For the standard deviation in Figures 2 and 6, it is not clear whether it is calculated from each individual model first, then averaged across models or it is the across model standard deviation.

The standard deviation presented is the across-model standard deviation and is an attempt to show the intermodel spread. We have specified this at several places in the revised text - including the figure captions.

Also, it seems that the standard deviation is not discussed in any detail in the manuscript. If so, those panels could be removed 40 from Figures 2 and 6.

Whilst we have not explicitly mentioned these fields, they are often alluded to in our discussion of what changes are 'consistent' or 'robust'. We feel they are important to provide to future researchers using the ensemble, so hope to keep these figure panels.

**Reviewer 2**

35

45 This manuscript presents the very recent simulations performed within the PMIP4-CMIP6 project and aiming at reproducing the climate of the mi-Holocene (6000 years ago). This climate was characterized by specific external forcing (greenhouse gases concentrations) and orbital parameters, which have been prescribed in a consistent way in the different models, following a previous study (Otto-Bliesner et al. 2017). At the time of the publication, twelve models have performed these simulations and are analyzed here, which is equivalent to the number of models having performed the previous coordinated exercise. The

50 manuscript proposes a first analysis of four specific features of the midHolocene climate: surface temperature, monsoons, extratropical hydrological responses and the ocean circulation. A final section discusses general assessments of the midHolocene climate.

The paper is in general very well written and presented, and figures are of high quality. References are thoroughly cited. This paper is clearly an important milestone for the PMIP4-CMIP6 intercomparison process, benchmarking the publication of

55 the simulations and probably calling for subsequent more original analysis. I only have a few major comments and a series of minor ones listed below. I want to indicate to the authors that I am not a paleo-expert, I come from the climate modeling community, typically rather focusing and the historical and future periods.

We thank you for your diligent review and hope that the revisions we've made resolve your concerns.

**60 Main comments**

On the calendar adjustment (section 2.2), I am surprised that this calendar adjustment has to be performed offline. Is there a fundamental reason that the protocol does not include a change of the online calendar? It is now obvious that our discussion of the calendar impact was unclear, so we have rephrased several sentences in this subsection. The insolation changes resulting from the altered orbital configuration are a key part of the experiment protocol. The problem here is to do with the aggregating

65 of output data during run time up to monthly resolution. To fix this online can require substantial modification of a model's output processing code, which would act as hurdle to participation.

Could you provide a comparison of the effect of the calendar adjustment for a variable for which it is proven useful (temperature?). Here, you only give numbers for the monsoon system, which you claim is not suited for this adjustment.

- 70 This form of calendar adjustment has never previously been implemented in a multi-model study, despite several calls for it (e.g. Kutzbach and Gallimore, 1988; Joussaume and Braconnot, 1997; Bartlein and Shafer, 2019). It is generally considered a minor error. The creation of easy-to-use software by Bartlein and Shafer (2019) has meant it has been possible to include it for the first time here. Justifications and use cases are provided by Bartlein and Shafer (2019).
- 75 It sounds illogical to me to assess (in this case to reject) the reliability of the calendar adjustment for the monsoon system using a fixed domain (lines 95-98) and just after (lines 99 and following) explain that the spatial extent of the monsoon varies so that the latter has to be adjusted for future assessments. Could you justify or modify? Thank you for pointing out the apparent contradiction raised by this juxtaposition. We have altered the order of these two
- subsections. Unfortunately the number of gridboxes meeting the criteria to be considered within the *midHolocene* monsoon
  domain is different between the original and calendar-adjusted monthly resolution output. And neither are completely identical to the number of gridboxes meeting the criteria when using the daily-resolution data. We therefore had little option but to only consider include in our average gridboxes that are within the domain in all three instances.

**Section 3.5 is relatively confusing.**

85 We agree that this section was not sufficiently clear and have rewritten it. We have substantially revised the first paragraph in this subsection, which hopefully makes its clearer.

L. 304-307: I have to say that I don't understand the point that is made here. First, grammatically, I don't understand the "they" 1. 305. And logically, I don't understand the link of what is said here with the beginning of the paragraph.

**90 We hope that the substantially revised and expanded paragraphs more understandable.**

L. 309 "noisy" would be more accurate than "chaotic" I think.

We agree that chaotic has a more specific meaning in the context of climate. We think it is better to describe the distribution of significant points as "random" and we have modified the sentence accordingly. Note we have modified the paragraph slightly

95 to remove the implied double negative, and now say that "There are hardly any locations that exceed the false discovery rate"

Paragraph beginning 1. 320: I find this paragraph difficult to follow and perhaps not so well organized.

We have rewritten this and split it into two paragraphs, one dealing with the change in sensitivity between model generations, and one exploring the possible link between sensitivity and seasonality. Please see specific comments below.

100

105

L. 322-323: it is not clear to me how Fig. 1 shows that the protocol is responsible for the detected differences.

Our argument rests on two pieces of evidence. The first is that although some models show higher sensitivity, there is no real difference in the range of sensitivities shown by the two ensembles. The second point is that the change in GHG forcing, and specifically the lowering of  $CO_2$  by 20 ppm in the CMIP6/PMIP4 simulations is consistent with the observed cooling. There is a strong correlation between the simulated cooling shown in panel (e) and the implied change due to this change in forcing

shown in panel (f). We now are more explicit about these arguments in the text.

The fact that "there is no inherent relationship between climate sensitivity and seasonality is repeated twice (l. 324 and l. 326) with no obvious demonstration in-between.

- 110 Our intention here was to state that there is no reason to expect a relationship over the oceans but that we might expect a relationship over land because of the feedbacks. Our, as yet limited analyses, do not support the idea of a relationship over land but we do not want to rule out the possibility that such a relationship might be found and we do want to encourage people to investigate this. We have rewritten this paragraph to make the argument clearer.
- 115 It could/should be said more explicitly that Fig. 11 is just an example and that checks have been performed for other features of the miHolocene climate features (have they?)

We now state that this is an example situation, although a telling one. We have performed similar checks on the other features listed in the Tab. S1 without showing significant correlations. However, this is only a limited subset of regions that have typically been used for data-model comparison, so it is still possible that such a relationship exists. We have rewritten the

120 paragraph to make it clearer.

I don't fully understand the last sentence of the paragraph: in what sense could it provide a constrain then? We want to say that our analysis of a limited number of examples should not dissuade future work on the role of climate sensitivity in the mid-Holocene simulations. We have rephrased the sentence.

125

**Minor comments**

1. 129: I don't know what recrine (212) is, please explain.

This was a typo for marine and we have corrected this in the revised text. Our intention here is to indicate the number of marine versus terrestrial records available

130

lines 35-38: the increased climate sensitivity in the CMIP6 generation of climate models is still under investigation and I think it is worth specifying: add "The reasons for this increase is still under investigation, but it may influence also the sensitivity of the models to the midHolocene external forcing."

We agree that it is worth adding something on this, but we have additionally clarified that some models have increased sensitivity and some decreased sensitivity.

End of section 2.1: on the specificities of the protocol and the models differences, I would like to advertise the site https://esdoc.org/ that provides a detailed documentation of all these points, with a full documentation of each model.

A shout out to ES-DOC has now been inserted. We have also included a new supplementary table (S1) to provide the doi for each of the simulations (as recommended by CMIP/ESGF).

L. 138: as 'as compared to piControl conditions' after 'Europe' and specify Fig. 1a. We have made this clarification.

L. 155-156: From Fig. 1, I am not completely sure that I can state that "the change in MAT with respect to the piControl in the PIMP4-CMIP6 ensemble is less than in the PMIP3-CMIP5: I see indeed a weaker warming at high latitudes but a stronger cooling in the tropics. Please correct or clarify.

We had intended this sentence to incorporate the direction, rather than just magnitude of the changes. We have altered 'less' to 'generally cooler' to remove this ambiguity.

150

L. 166: I would suggest naming explicitly the C20 reanalysis in the brackets. This has been done

L. 170-171: as shown in Fig. 1(e), the PIMP4-CMIP6 models are generally colder than the PMIP3-CMIP5 ones. The reduced
warm bias along the EBUS could simply be a consequence of this. Thorough analysis of the evolution of this bias would require the use of relative temperature (e.g. Hourdin et al. 2015, GRL).

We agree about this observation about the piControl biases. We have amended the sentence to point out it that it doesn't

necessarily mean the models are better. We note that Fig 1e shows the difference in the midHolocene signal though, so actually the evidence behind this statement is not presented anywhere in the manuscript. It will be available in IPCC AR6.

160

L. 175: I don't understand how this correlation coefficient is computed.

It was a correlation coefficient between the Arctic dots in panels A and B. In fact, we could make our point just as easily without such quantification. The sentence now reads: "There is no simple relationship between a model's representation of the preindustrial temperature and the magnitude of its simulated mid-Holocene temperature response (Fig. 4)"

165

L. 190: I would rather use the term "intermodel spread", as in several other places of the manuscript. Standard deviation typically relates to temporality in my view. Changes in the legend of Fig. 1, 2 and 6 are also needed. This is a useful suggestion and we have implemented it throughout the manuscript.

L. 233-235: the term "the changes of precipitation" is sometimes confusing: I think you speak of the changes of each PMIP exercise with respect to piControl, but one could also read as the changes from PMIP3 to PMIP4. I suggest clarifying. We have rephrased two sentences here. They now read "However, there is little relationship between the *piControl* precipitation biases and the simulated midHolocene changes in precipitation (Fig. S1). The variations in the midHolocene rainfall signal appear to be more related to monsoon dynamics rather than orbitally-induced local temperature variations."

175

L. 239: remove the word "indeed"

It has been removed.

L. 250 this sentence does not mean much to me. Large spread with respect to what?

180 We have expanded this phrase. It now reads "There are large differences in the simulated change in mid-Holocene precipitation between different models, as shown by the standard deviation around the ensemble mean, in both the PMIP4-CMIP6 and PMIP3-CMIP5 ensembles (Fig. 6 & 8). Unsurprisingly, the largest differences between models occurs where the simulated change in precipitation is also largest (Fig. 6)."

185 L. 277 I would rather write "a decline of up to  $\sim 15\%$ " This has been implemented.

L. 280: It should be made clear that the magnitude of the AMOC computed at 50°N in z-coordinates is a little bit misleading, since in fact intense transport of water mass occurs at similar depths, but with very different densities (e.g. Zhang GRL 2010). We recognise this possible confounding factor, but feel such a technical caveat is not helpful in this situation. Our purpose was

190

to not rely only on 30°N - which seemed justified given Reviewer's 3 comments about its specification. Instead we have added a sentence acknowledging potential issues, but without specificyig what they are in detail. The manuscript now reads "Using a single metric to categorise AMOC is awkward – that two measures, both with their own foibles, support show the same result 195 L. 286: I don't clearly understand the link that is made with the magnitude of low frequency internal variability in AMOC. Clarify.

We had meant to suggest that the changes in the AMOC shown may be within the bounds of natural variability. As the sentence caused confusion and did not add much to the narrative, it has been removed.

200 Conclusion section will have to be changed accordingly to previous remarks (on section 3.5 in particular) We have revised several sentences in the conclusions section. As neither the methodological revisions nor the increase in ensemble size fundamentally revised our findings, these are relatively minor.

L. 365: I have no proof at this stage that the changes in the implementation of aerosols in CMIP6 is an improvement. Wedid not either, and this sentence was written in anticipation that some relevant work would be submitted at the same time as this manuscript. We have changed 'improvement' to 'advances', which does not specify that CMIP6 is better than CMIP5 so forthrightly. We have also weakened the phrasing around the aerosols.

**Reviewer 3**

210 The manuscript presents the recent simulations of mid-Holocene performed within the PMIP4-CMIP6. It is a valuable and interesting work as it will be one of the main references for the future studies. The text is written and structured well, and the storyline is what would be expected from such a paper but I have few main critical and some major/minor comments as are listed below. Overall, I think it should be published in Climate of the Past, after a minor-moderate revision.

We thank the reviewer for their kind comments, and hope that the revisions we have implemented are sufficient to satisfy their expectations.

**Main comments**

As one of the important components of the climate, I wonder why the sea ice story was not included except the short sentence at line 193. I suggest to add the 2-D maps of sea ice concentration in the Arctic (for both summer and winter) which would also be relevant to the AMOC story.

```
also be relevant to the AMOC story.
```

There are two main reasons that we did not include sea ice in this manuscript. Firstly, we wanted to constrain the scope of this manuscript to a manageable amount of analyses as sea ice analyses can easily become stand alone manuscripts (e.g. Berger et al., 2013; Kageyama et al., 2020). Secondly, there were technical issues around the calendar adjustment using the Bartlein and Shafer (2019) software. It had been developed and evaluated on surface temperature and precipitation. The software

225 development required for it to adjust fields on rotated grids has been attempted, but not been scientifically validated for sea ice coverage. Using one of the three of the models that have provided daily fields, we have performed that.

We have now performed a substantial amount of analysis into the sea ice coverage changes at the midHolocene, which have culminated in the addition of a new figure. Despite our efforts, we have not been able to find a visualisation method using the 2-D maps suggested by the reviewer that adds to narrative. We therefore include two summary scatter plots, inspired by Berger et al. (2013), that try to provide a concise summary of the results.

230

240

For instance, when there is sea ice covering part of the deep water formation (DWF) region in the Labrador Sea (due to model bias), DWF is reduced. If the model has too much sea ice over the Labrador Sea in both PI and midHolocene, then that can partially explain why the AMOC does not differ much between the two periods, as any freshening or cooling cannot influence the DWF.

We feel that this particular aspect needs to be addressed first by authors evaluating the performance of the sea ice models against observations in the historical simulations. Several works may have been submitted on the topic, but there is disappointingly little openly accessibly at the moment for us to base our derived analysis upon. We do now highlight PMIP3 previous work on the topic (Găinuşă-Bogdan et al., 2020) in a hope to inspire future researchers.

With regards to AMOC, it would be nice to say something (few sentences) about the regions where deep water formation happens based on mixed layer depth values in March. It could be that if there is sea ice over the Labrador (recalling my previous comment), then the region of deep water formation might shift.

245 Detailed, subsequent analysis of the mid-Holocene AMOC and deep water formation has been initiated, but is the focus of a future paper.

Again about AMOC, I know that normally 30° N (models) or 26.5° N (observations) are chosen for calculating the maximum AMOC value. Since this latitude of maximum AMOC can differ between models as well as the two periods of study, I would

not pick one latitudinal point. Instead, I will define a range e.g., 25-35° N to calculate the maximum AMOC.
 We understand the reviewers position, but would prefer to stick with the definition adopted by the IPCC'S 6th Assessment Report.

I also suggest to add the observations to your plot (RAPID-MOCHA array observations, Smeed et al., 2017).

255 This is a sensible suggestion. We have now added the RAPID-MOCHA array to compare with 30°N, and also the OSNAP array to provide a more northerly benchmark for the 50°N metric.

**Specific comments regarding the text and analysis**

Methods: The mean values, are they averaged over the entire simulated years mentioned in Table 1?

260 Yes. We have now added a footnote to the table to clarify this.

Line 86: Can you say in one sentence how is PaleoCalAdjust performing in general?

Bartlein and Shafer (2019) provide convincing evidence that it is performing well on PMIP3 mid-Holocene simulations for monthly surface temperature and precipitation. We have now mentioned this in the revised text through an additional sentence: "This software was developed and been favorably evaluated for monthly temperature and precipitation variations with both

PMIP3-CMIP5 and transient simulations (Bartlein and Shafer, 2019)."

Lines 88-98: For the annual mean I understand you do not need calendar adjustment. But if you use your daily values from PaleoCalAdjust and make the annual mean, how much would it differ from the main annual mean? This can give you some

270 ideas about the potential interpolation errors (if there is no original daily data). Thank you for your suggestion. A subset of the models have actually provided simulation output at a daily resolution, so this step is not necessary. We provided an example of this kind of analysis for Arctic sea ice extent as a supplement in our earlier Author Comment. We do not feel this manuscript is the correct place to provide additional evaluation of the methodlogical technique. Your comments provide further motivation for a further detail analysis to be published.

275

265

Line 98: I do not understand "we have therefore..." so you use the method when you think it is good?

Fundamentally, yes. There is a balance to navigate between the original size of the error due to the calendar misalignment and errors introduced by PaleoCalAdjust from the interpolation step. The analysis in the previous few sentences demonstrates that the interpolation errors are greater than the misalignment, so there is little advantage using PaleoCalAdjust for this diagnostic.

280

Line 107: in this line and any other lines (line 219) please change "interannual variability" to internal climate variability because the variability is not only interannual.

This new terminology has been adopted in the text. It is however retained in the caption, because it concisely conveys the message that the standard deviations is measured across a time series that is only resolved annually.

285

Lines 154-161: move these lines to after line 140.

This restructuring of the paragraphs has been adopted, although the one sentence about seasonal changes is moved instead to the end of the paragraph describing Fig. 2.

290 Line 171: "...colder conditions over the Labrador current..." which figure you are referring to? And I assume you meant Labrador Sea and not current?

**We have corrected 'current' to 'Sea' and added a reference to Fig. 3b.**

Line 174: not only in the tropics but over the oceans in general there is a better match

295 We have now added this insight.

Line 234: "... change in precipitation" change between what? This has actually been altered to 'rainfall changes', to stop the paragraph feeling too repetitive (as per next comment).

300 Line 234-236: you used "change" three times in one sentence, modify please and combine it with the previous sentence. This sentence has now been modified to only include 1 'change'. We have not combined with the previous sentence though, as it's already rather long.

Line 324: role

305 We have implemented this change.

Line 374: "... need for improved physics and processes..." This change has been implemented.

Figure 3 caption: check the years where observation was used e.g., you wrote "1981-1900". Also check the rest of caption. This should have been 1871-1900, and has now been corrected.

Figure 5: would it be possible to make the similar figure for the observation/reanalysis?

It is possible, and was included in the supplement to our earlier Author Comment. However, given that the boundary of the 315 domain in the observations/reanalysis is marked both panels already, we feel it is superfluous to include an additional panel in the manuscript.

[revised manuscript text omitted]

---

## Author Response (AR2)

**Reply to the editor's comments: Large-scale features and evaluation of the PMIP4-CMIP6 midHolocene simulations (cp-2019-168)**

C. M. Brierley *et al.*

**Correspondence:** c.brierley@ucl.ac.uk

**Summary of changes**

We would like to thank the Editor for accepting the manuscript pending some very minor changes. In light of the two errors that you spotted in reading it, we have given the manuscript a thorough proof-read and implemented a variety of corrections. Many of them are typographical and so have not been documented here, but are visible in the versions of the manuscript with tracked changes below. However, we would like to highlight a few revisions worth noting.

1. The editor is correct that we did mean Fig. 11 on line 19 of the supplement. This has been altered.

2. In working on the *lig127k* companion paper (Otto-Bliesner et al., 2020), the issue which had prevented FGOALS-f3-L and FGOALS-g3 being plotted on Fig. 10 was resolved. They are now plotted in this figure.

3. The vague sentence (around L38) about climate sensitivity has been revised to mentioned the recently published work by Zelinka et al. (2020).

4. Palaeoclimate has been spelt consistently throughout

5. An erroneous reference to Fig S3 has been altered to direct readers to Fig S1 (on L253)

6. A sentence discussing the CMIP6 projections for AMOC has been in added in light of the publication of Weijer et al. (2020).

7. Instances where CMIP came before PMIP in ensemble names have been harmonised.

8. A sentence comparing the *midHolocene* to *lig127k* Arctic sea ice responses has been introduced.

9. The discussion about ensemble that not show a 'higher ECS gives higher seasonality change' in Fig. 12 has been improved on L377.

10. The sentence about MIROC-ES2L getting the reconstructed damping of ENSO has been revised for greater clarity.

11. The erroneous reference to Tab. S4 in the caption of Fig. 4 has been corrected now point to Tab. S3.

12. After submission of Ohgaito et al. (2020) to GMD, it has been cited in Tab. 1.

13. The model acronym for the Hadley Centre model now has the -LL flag added to it in Tab. 1 – so it now appears as it does on the ESGF.

14. The ESGF references for the AWI-ESM-1-1-LR *midHolocene* simulation is now available and has been added to Table S1 (Shi et al., 2020).

**References**

Ohgaito, R., Yamamoto, A., Hajima, T., O'ishi, R., Abe, M., Tatebe, H., Abe-Ouchi, A., and Kawamiya, M.: PMIP4 experiments using MIROC-ES2L Earth System Model, Geoscientific Model Development Discussions, 2020, 1–29, https://doi.org/10.5194/gmd-2020-64, https://gmd.copernicus.org/preprints/gmd-2020-64/, 2020.

Otto-Bliesner, B. L., Brady, E. C., Zhao, A., Brierley, C., Axford, Y., Capron, E., Govin, A., Hoffman, J., Isaacs, E., Kageyama, M., Scussolini, P., Tzedakis, P. C., Williams, C., Wolff, E., Abe-Ouchi, A., Braconnot, P., Ramos Buarque, S., Cao, J., de Vernal, A., Guarino, M. V., Guo, C., LeGrande, A. N., Lohmann, G., Meissner, K., Menviel, L., Nisancioglu, K., O'ishi, R., Salas Y Melia, D., Shi, X., Sicard, M., Sime, L., Tomas, R., Volodin, E., Yeung, N., Zhang, Q., Zhang, Z., and Zheng, W.: Large-scale features of Last Interglacial climate: Results from evaluating the *lig127k* simulations for CMIP6-PMIP4, Climate of the Past Discussions, 2020, 1–41, https://doi.org/10.5194/cp-2019-174, https://cp.copernicus.org/preprints/cp-2019-174/, 2020.

Shi, X., Yang, H., Danek, C., and Lohmann, G.: AWI AWI-ESM1.1LR model output prepared for CMIP6 PMIP midHolocene, Earth System Grid Federation, https://doi.org/10.22033/ESGF/CMIP6.9332, 2020.

[revised manuscript text omitted]

**Simulated Temperature Changes**. The surface air temperature changes averaged in 30°latitude-wide bands are computed for every model included in the study. These changes are computed separately over the ocean and land as well. The annual mean SST change should closely track the surface air temperature change presented here, but can vary in regions of sea ice cover. A common land sea mask at $1° \times 1°$ resolution is used for all models (Phillips et al., 2014). The relative weightings of land, sea and combined areas are provided to allow averages over other regions to be determined. **This table is provided as a file called `PMIP4-midHolocene-latband-tempchange-table.xls.`**

**Table S1.** Digital Object Identifier (doi) for each simulation from CMIP6 and CMIP5. Should the hyperlinks in the table not work, the web address can be created manually by adding `https://dx.doi.org/` in front of each doi. The full citations are in the References.

| model | *midHolocene* | *piControl* |
|---|---|---|
| AWI-ESM-1-1-LR |  10.22033/ESGF/CMIP6.9332 | 10.22033/ESGF/CMIP6.9335 |
| CESM2 | 10.22033/ESGF/CMIP6.7674 | 10.22033/ESGF/CMIP6.7733 |
| EC-Earth3-LR | 10.22033/ESGF/CMIP6.4847 | 10.22033/ESGF/CMIP6.4801 |
| FGOALS-f3-L | 10.22033/ESGF/CMIP6.12014 | 10.22033/ESGF/CMIP6.3447 |
| FGOALS-g3 | 10.22033/ESGF/CMIP6.3409 | 10.22033/ESGF/CMIP6.3448 |
| GISS-E2-1-G | 10.22033/ESGF/CMIP6.7225 | 10.22033/ESGF/CMIP6.7380 |
| HadGEM3-GC31-LL | N/A | 10.22033/ESGF/CMIP6.6294 |
| INM-CM4-8 | 10.22033/ESGF/CMIP6.5077 | 10.22033/ESGF/CMIP6.5080 |
| IPSL-CM6A-LR | 10.22033/ESGF/CMIP6.5229 | 10.22033/ESGF/CMIP6.5251 |
| MIROC-ES2L | 10.22033/ESGF/CMIP6.5646 | 10.22033/ESGF/CMIP6.5710 |
| MRI-ESM2 | 10.22033/ESGF/CMIP6.6860 | 10.22033/ESGF/CMIP6.6900 |
| NESM3 | 10.22033/ESGF/CMIP6.8773 | 10.22033/ESGF/CMIP6.8776 |
| NorESM1-F | 10.22033/ESGF/CMIP6.11591 | 10.22033/ESGF/CMIP6.11595 |
| NorESM2-LM | 10.22033/ESGF/CMIP6.8079 | 10.22033/ESGF/CMIP6.8217 |
| UofT-CCSM-4 | N/A | N/A |
| bcc-csm1-1 | 10.1594/WDCC/CMIP5.BCB1mh | 10.1594/WDCC/CMIP5.BCB1pc |
| CCSM4 | 10.1594/WDCC/CMIP5.NRS4mh | 10.1594/WDCC/CMIP5.NRS4pc |
| CNRM-CM5 | 10.1594/WDCC/CMIP5.CEC5mh | 10.1594/WDCC/CMIP5.CEC5pc |
| CSIRO-MK3-6-0 | 10.1594/WDCC/CMIP5.CQMKmh | 10.1594/WDCC/CMIP5.CQMKpc |
| CSIRO-MK3L-1-2 | N/A | N/A |
| EC-Earth-2-2 | N/A | N/A |
| FGOALS-G2 | 10.1594/WDCC/CMIP5.LSF2mh | 10.1594/WDCC/CMIP5.LSF2pc |
| FGOALS-S2 | 10.1594/WDCC/CMIP5.LIFSmh | 10.1594/WDCC/CMIP5.LIFSpc |
| GISS-E2-R | 10.1594/WDCC/CMIP5.GIGRmh | 10.1594/WDCC/CMIP5.GIGRpc |
| HadGEM2-CC | 10.1594/WDCC/CMIP5.MOGCmh | 10.1594/WDCC/CMIP5.MOGCpc |
| HadGEM2-ES | 10.1594/WDCC/CMIP5.MOGEmh | 10.1594/WDCC/CMIP5.MOGEpc |
| IPSL-CM5A-LR | 10.1594/WDCC/CMIP5.IPILmh | 10.1594/WDCC/CMIP5.IPILpc |
| MIROC-ESM | 10.1594/WDCC/CMIP5.MIMEmh | 10.1594/WDCC/CMIP5.MIMEpc |
| MPI-ESM-P | 10.1594/WDCC/CMIP5.MXEPmh | 10.1594/WDCC/CMIP5.MXEPpc |
| MRI-CGCM3 | 10.1594/WDCC/CMIP5.MRMCmh | 10.1594/WDCC/CMIP5.MRMCpc |

N/A indicates that a doi is not available.

**Table S2. Key metrics of change in the PMIP4-CMIP6 *midHolocene* simulations** *(see above for further details)*

| | Global mean temperature (°C) | Summer NH high-lat. land (°C) | Drier Eastern North America (mm/yr) | Midcontinental Eurasia rainfall (mm/yr) | Midcontinental Eurasia Seasonality (°C) | Central Asian Seasonality (°C) | N. African monsoon expansion (°N) | Drier South America (mm/yr) | Indo-Gangetic rainfall (mm/yr) | Niño3.4 Variance‡ (%) | p(suppressed ENSO) in piControl§ (%) | p(suppressed ENSO) in midHolocene§ (%) | Eq. Pac SST gradient‡ (%) |
|---|---|---|---|---|---|---|---|---|---|---|---|---|---|
| | | | *Extratropical* | | | | | *Tropical* | | | | | |
| AWI-ESM-1-1-LR | -0.4 | 0.0 | -58 | -21 | 2.6 | 2.9 | 3.1 | 68 | 115 | -41 | – | – | -8 |
| CESM2 | -0.2 | 0.7 | -54 | -16 | 2.8 | 3.1 | -0.2 | -97 | 125 | -16 | 2.4 | 5.7 | -7 |
| EC-Earth3-LR | -0.1 | 1.8 | -28 | 12 | 2.3 | 2.3 | -0.5 | -29 | 166 | -31 | – | – | -5 |
| FGOALS-f3-L | -0.4 | 0.6 | -24 | -11 | 3.0 | 3.0 | 1.5 | -85 | 165 | 4 | 2.8 | 1.2 | 0 |
| FGOALS-g3 | -0.2 | 1.1 | -92 | -58 | 4.2 | 4.1 | 1.8 | -258 | 57 | -14 | 0.2 | 2.5 | -1 |
| GISS-E2-1-G | -0.4 | 0.7 | -15 | -9 | 2.4 | 2.6 | 1.6 | -60 | 188 | 2 | 1.8 | 5.6 | 4 |
| HadGEM3-GC31-LL | -0.1 | 1.2 | -9 | 2 | 3.0 | 3.8 | 2.3 | -102 | 207 | -8 | 0.6 | 0.6 | -3 |
| INM-CM4-8 | -0.3 | 0.7 | 24 | -2 | 2.7 | 3.1 | 2.1 | -96 | 212 | 7 | 1.5 | 14.2 | -4 |
| IPSL-CM6A-LR | -0.4 | 0.5 | -23 | -32 | 3.5 | 3.0 | 0.9 | -72 | 160 | -13 | 1.7 | 3.4 | -3 |
| MIROC-ES2L | -0.5 | 0.6 | 30 | -26 | 2.8 | 3.4 | 1.3 | -111 | 77 | -49 | 7.3 | 82.4 | 16 |
| MPI-ESM1-2-LR | -0.4 | 0.6 | -26 | -18 | 2.8 | 3.0 | 3.7 | -179 | 189 | -28 | 1.1 | 7.4 | -4 |
| MRI-ESM2-0 | -0.2 | 0.7 | -22 | -15 | 2.5 | 2.7 | 3.3 | -179 | 189 | -36 | 4.8 | 34.5 | 0 |
| NESM3 | -0.3 | 0.9 | 59 | 24 | 2.6 | 2.5 | 3.1 | -155 | 177 | -24 | 2.1 | 5.2 | -4 |
| NorESM1-F | -0.4 | 0.4 | -6 | -8 | 3.4 | 3.6 | 1.4 | -116 | 158 | -6 | – | – | -6 |
| NorESM2-LM | -0.2 | 0.5 | 137 | 137 | 3.3 | 3.0 | -1.9 | -85 | 255 | 11 | – | – | -8 |
| UofT-CCSM-4 | -0.2 | 1.1 | -8 | -3 | 3.1 | 2.8 | 1.9 | -117 | 114 | -48 | – | – | -2 |
| Reconstructed | 0.5† | 0.7* | -93¶ | 121¶ | – | – | – | – | – | – | – | – | – |
| PMIP4 Average | -0.3 | 0.8 | -7 | -3 | 2.9 | 3.1 | 1.7 | -99 | 162 | -18 | 2.4 | 14.8 | -1.9 |
| PMIP3 Average | -0.1 | 1. | -10 | -4 | 2.6 | 2.9 | 3.0 | -83 | 175 | -11 | 3.7§ | 5.8§ | -3 |
| PMIP3 Spread | 0.2 | 0.5 | 19 | 15 | 0.4 | 0.4 | 4.2 | 46 | 81 | 14 | 3.2§ | 4.3§ | 6 |

†Median reconstructed global mean value from Kaufman et al. (2020a), with 80% confidence interval of 0.3–0.9 °C. *average of the difference in summer and winter reconstructions within the region from Kaufman et al. (2020b) compilation. ¶average of reconstructions within the region from Bartlein et al. (2011) compilation. ‡Values published in Brown et al. (submitted). §Using the analysis approach of Emile-Geay et al. (2016) with PMIP3 values directly from it.

[Figure]

**Figure S1. Simulated North African monsoon through multiple phases of PMIP-CMIP.** (top panel) Biome distributions (desert, steppe, xerophytic and savannah/dry tropical forest) as a function of latitude for present (red circles) and 6 ka (green triangles), showing that steppe vegetation replaces desert at 6 ka as far north as 23°N (middle panel) Annual mean precipitation changes (mm/yr) over Africa (20°W–30°E) for the Mid-Holocene climate across multiple PMIP generations. The black hatched lines are estimated upper and lower bounds for the additional precipitation required to support steppe at each latitude during the mid-Holocene based on water-balance modelling and the modern climatic requirements for desert and grassland plants. (bottom panel) The rainfall distribution in piControl simulations for each model. Three different observationally-based datasets are shown in black: GPCP (Adler et al., 2003), CMAP (Xie and Arkin, 1997), and CRU (New et al., 2000). (Adapted from Joussaume et al., 1999; Braconnot et al., 2007, 2012)

[Figure]

**Figure S2. Statistical description of site-level comparison of simulated mid-Holocene climate changes to reconstructions.** The performance of both the CMIP6 and CMIP5 ensembles are assessed by comparing the annual mean temperature changes and difference between summer mean temperature changes and winter mean temperature changes to multi-proxy Temperature 12k database (red, green; Kaufman et al., 2020b) and mean annual precipitation and difference between mean temperature of the warmest month (MTWA) changes and mean temperature of the coldest month (MTCO) changes to the pollen-based reconstructions (yellow, blue, purple; Bartlein et al., 2011). The better a model's changes fit with the reconstructions, then closer it should be to the green square (Taylor, 2001). The correlation coefficient is plotted on the azimuth, and the radial distance presents the ratio of the standard deviation in the model and reconstructions  (after adjustment to account for the existence of uncertainty in them, Hargreaves et al., 2013).

[Figure]

**Figure S3. Alternate presentation of the data-model comparison**. Regional comparisons using Monte-Carlo sampling of both the reconstruction uncertainty (Bartlein et al., 2011) and model uncertainty as expressed by interannual variability at individual proxy locations. The regions are defined as Europe (35–70°N, 10°W–30°E), West Africa (0–30°N,30°W–30°E) and North America (20–50°N,140–60°W).

**Note: Table S3 is provided as an external spreadsheet called `PMIP4-midHolocene-latband-tempchange-table.xls`**